# Regulation of coordinated muscular relaxation in *Drosophila* larvae by a pattern-regulating intersegmental circuit

Atsuki Hiramoto [1], Julius Jonaitis[2], Sawako Niki[1,3], Hiroshi Kohsaka [1], Richard D. Fetter [4], Albert Cardona [4,5,6], Stefan R. Pulver[2] & Akinao Nose [1,7✉]

Typical patterned movements in animals are achieved through combinations of contraction and delayed relaxation of groups of muscles. However, how intersegmentally coordinated patterns of muscular relaxation are regulated by the neural circuits remains poorly understood. Here, we identify Canon, a class of higher-order premotor interneurons, that regulates muscular relaxation during backward locomotion of *Drosophila* larvae. Canon neurons are cholinergic interneurons present in each abdominal neuromere and show wave-like activity during fictive backward locomotion. Optogenetic activation of Canon neurons induces relaxation of body wall muscles, whereas inhibition of these neurons disrupts timely muscle relaxation. Canon neurons provide excitatory outputs to inhibitory premotor interneurons. Canon neurons also connect with each other to form an intersegmental circuit and regulate their own wave-like activities. Thus, our results demonstrate how coordinated muscle relaxation can be realized by an intersegmental circuit that regulates its own patterned activity and sequentially terminates motor activities along the anterior-posterior axis.

[1] Department of Complexity Science and Engineering, Graduate School of Frontier Sciences, The University of Tokyo, Chiba, Japan. [2] School of Psychology and Neuroscience, University of St Andrews, St Andrews, UK. [3] Research Center for Advanced Science and Technology, The University of Tokyo, Tokyo, Japan. [4] HHMI Janelia Research Campus, Ashburn, VA, USA. [5] Department of Physiology, Development and Neuroscience, University of Cambridge, Cambridge, UK. [6] MRC Laboratory of Molecular Biology, Cambridge, UK. [7] Department of Physics, Graduate School of Science, The University of Tokyo, Tokyo, Japan. ✉email: nose@k.u-tokyo.ac.jp

One of the major goals in neuroscience is to understand how neural circuits regulate movements. Animal movements are generated by sequential contraction and relaxation of muscles present in different parts of the body[1–3]. While muscle contraction provides the force for moving, timely muscular relaxation is also required for coordinated movements to occur[4–7]. For instance, while a particular muscle (e.g., a limb extensor) is contracted, its antagonistic muscle (e.g., a flexor) must be relaxed. Similarly, muscles in different parts of the body whose simultaneous contraction hinders appropriate body movements must be coordinately contracted and relaxed. Muscle relaxation is thought to be particularly important for fine control of movements required during playing sports and musical instruments[8]. Conversely, deficits in muscle relaxation have been associated with a wide spectrum of movement disorders such as myotonic dystrophy, Parkinson disease and dystonia[8–10].

Muscle contraction ends when transmission from motor neurons (MNs) is terminated, allowing muscle relaxation. Inhibitory motor neurons are present in some invertebrate species and play important roles in tuning muscle relaxation to meet behavioral demands[11–13]. However, termination of MN activity by upstream premotor circuits is also a major mechanism for regulating muscle relaxation. Previous studies in vertebrates and invertebrates have identified a number of premotor interneurons that play roles in patterning motor activities[1–3,14,15]. Although much is known about the timing of muscle contraction, such as how left-right or flexor-extensor alteration is regulated by the premotor circuits, less is known about the cellular and circuit mechanisms of interneuron coordinated muscle relaxation[6,7].

*Drosophila* larval locomotion is an excellent model for investigation of sensorimotor circuits at the single cell level[16–18] (Fig. 1a). The larva has a segmented body and normally moves by forward locomotion. However, when exposed to noxious stimulus to the head, it performs backward locomotion as an escape behavior[16,19,20]. Forward and backward locomotion are symmetric axial movements achieved by propagation of muscular contraction and relaxation along the body. In both forward and backward peristalses, contraction and relaxation of muscles occur in a segmentally coordinated manner, so as to generate a coherent behavioral output[21,22]. Recent studies identified command-like neurons, whose activation can elicit backward locomotion, including Wave and mooncrawler descending neurons (MDNs)[20,23]. Recent studies also identified several classes of premotor interneurons that regulate various aspects of larval peristaltic locomotion, including speed of peristaltic propagation, left–right coordination, and sequential contraction of antagonistic muscles[5,24–26]. One such class, the *period*-positive median segmental interneurons (PMSIs) are activated later than their target MNs and regulate the duration of MN activities, and thus could potentially play roles in the regulation of muscle relaxation. The activity of PMSIs and other inhibitory premotor interneurons is thought to be regulated by Ifb-Fwd and Ifb-Bwd neurons, during forward and backward locomotion, respectively[25]. However, none of the previously identified premotor neurons have been shown to be essential for normal patterns of muscle relaxation during peristaltic locomotion. Neither have any been shown to be essential for generating the propagation of axial activity.

Here, we report on the identification of a class of segmentally repeated cholinergic interneurons, that we have named Canon (cholinergic ascending neurons organizing their network) neurons, which regulate timely muscular relaxation during larval backward locomotion. Canon neurons show segmentally propagating activities during backward but not forward fictive locomotion at a timing much later than MNs. Optogenetic activation of Canon neurons induced relaxation of body wall muscles, whereas their genetic inhibition severely delayed muscle relaxation. Connectomics analysis (reconstruction of circuit structures in serial electron microscopy (EM) images[27]) revealed that Canon neurons send outputs to a large number of interneurons, but are particularly strongly connected to a group of first-order inhibitory premotor interneurons. Taken together, our results suggest that Canon neurons regulate muscle relaxation by providing delayed excitation to inhibitory premotor interneurons. We also found that Canon neurons in abdominal neuromeres connect with each other to form an intersegmental network, that regulates their own propagating activities. Thus, our results demonstrate how coordinated muscle relaxation can be regulated by the action of a pattern-regulating intersegmental circuit.

## Results

**Canon neurons are ascending interneurons active during backward but not forward locomotion.** Neurons that regulate specific motor patterns are often recruited in a manner related to the pattern. To identify neurons related to backward locomotion, we searched for interneurons showing backward-specific activity propagation by calcium imaging. We identified among the neurons targeted by *R91C05-Gal4* a class of segmentally repeated interneurons that we termed "Canon" due to their distinctive morphological and functional features. These neurons showed wave-like activity propagation in the backward direction but not in the forward direction (Fig. 1b, Supplementary Fig. 1a, b, and Supplementary Movie 1). Since *R91C05-Gal4* only targets Canon neurons in abdominal neuromeres A3–A5, we also generated an independent Gal4 line, *Canon-spGal4*, which drives expression in Canon neurons in A1–A6 neuromeres (Supplementary Fig. 1c). To study the timing of Canon neuron activity relative to MNs, we conducted simultaneous calcium imaging of Canon neurons and aCC MNs (Fig. 1c, d, Supplementary Fig. 1d, e, and Supplementary Movie 2)[28,29]. Canon neurons were activated after aCC MNs in the same neuromere and at a similar timing to aCC MNs in neuromeres 2–3 segments more anterior suggesting possible roles in the termination of segmental motor activity (Fig. 1d and Supplementary Fig. 1e). The calcium imaging also confirmed that Canon neurons were activated during fictive backward but not forward locomotion (Fig. 1c).

We next studied the neurite morphology and neurotransmitter phenotype of Canon neurons. We generated single-cell clones of Canon neurons using the MultiColor FLP-Out (MCFO) method[30], and studied axon and dendrite extension of individual Canon neurons (Fig. 1e, f). We also used the DenMark technique[31] to identify pre- (visualized with Syt-GFP) and post-synaptic (visualized with the DenMark marker) sites (Supplementary Fig. 1f). These analyses revealed characteristic morphological features of the Canon neurons, as detailed below. The cell bodies of Canon neurons were located in a ventrolateral region of the VNC (Fig. 1f), and extended an axon initially to the contralateral dorsal neuropil via the anterior commissure (Supplementary Fig. 1g), and then anteriorly to the T2 neuromere (Fig. 1e). The ascending axons sent collaterals in each of the two to three neuromeres anterior to the cell body and arborized presynaptic terminals in the dorsal neuropil, which are known to contain MN dendrites. Canon neurons expressed choline acetyltransferase (ChAT), a marker for cholinergic neurons, but not vesicular glutamate transporter (vGluT)[32], a marker for Glu, or GABA, (Fig. 1g and Supplementary Fig. 1h, i). The specificity of ChAT immunoreactivity was confirmed by the signal reduction upon ChAT-RNAi expression (Supplementary Fig. 1j, k). These results suggest that Canon neurons are cholinergic and most likely excitatory. To summarize, Canon neurons are segmental ascending interneurons activated after MNs during backward locomotion.

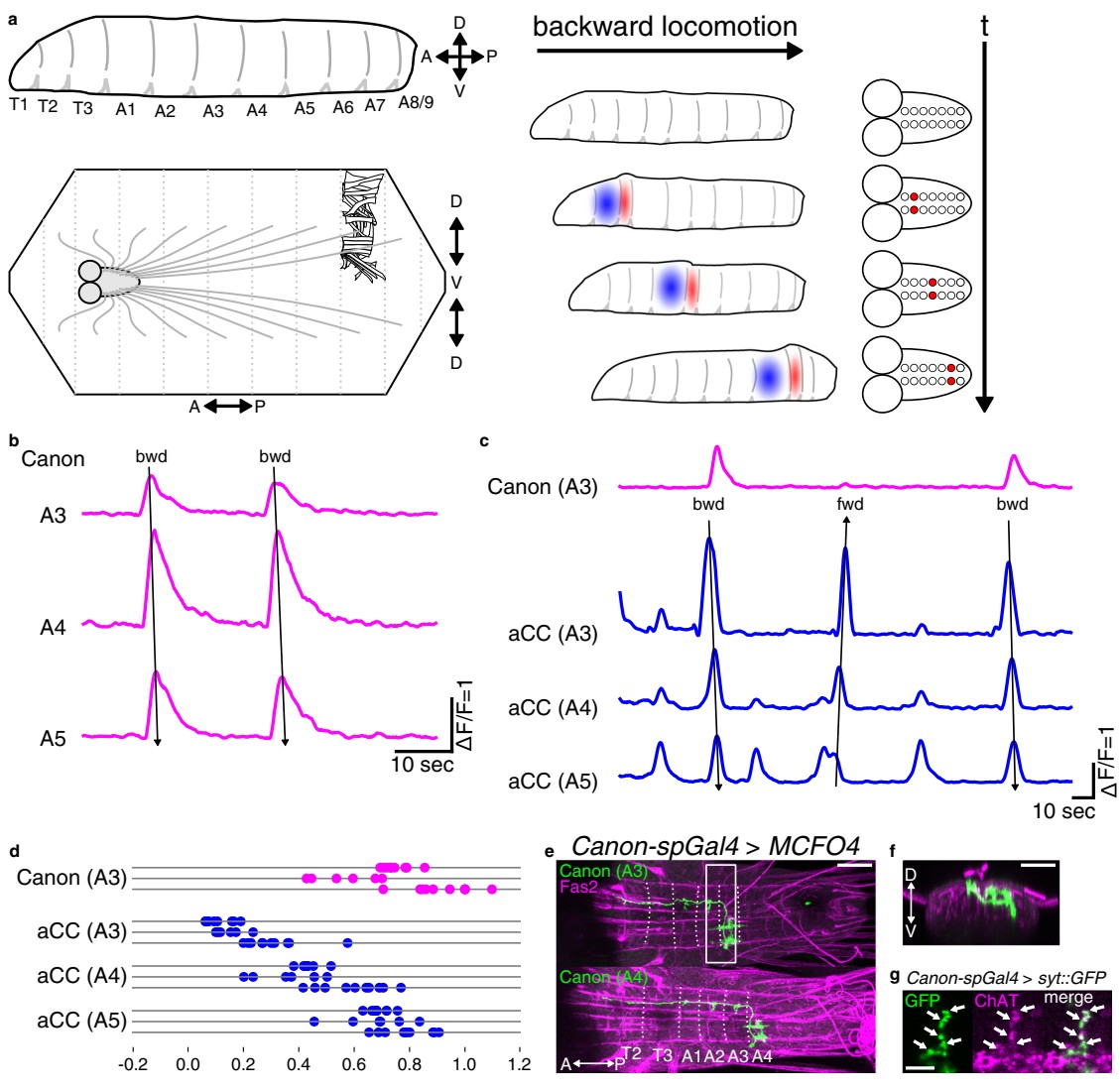

**Fig. 1 Activity and anatomy of Canon neurons. a** (Left) A scheme of a larval body which is segmented into T1–3 and A1–9 (top). A scheme of a dissected larva, body-wall muscles, and the CNS (bottom). A CNS is also segmented and MNs on located in each neuromere sends outputs to muscles located on corresponding segment. (Right) Larval peristaltic behavior is generated by the propagation of muscular contraction (red) and delayed relaxation (blue) (left), which is induced by propagation of activity of MNs (red) (right). **b** Canon neurons in A3–A5 neuromeres show propagating activity in the backward direction. **c** Simultaneous imaging of A3 Canon neuron (magenta) and A3–A5 aCC MNs (blue). Backward (bwd) and forward (fwd) fictive waves are indicated by arrows in **b**, **c**. **d** Linear phase plot of activity peaks in an A3 Canon neuron (magenta) and A3–A5 aCC MNs (blue) during backward waves. Each of the three horizontal lines represents an individual preparation. The timings of the activity peaks in each neuron relative to the phase of backward peristalses were plotted. Phase 0 is initiation of activity of A2 Canon neuron, while Phase 1 is peak of activity of A6 Canon neuron. $n = 26$ waves from three larvae. **e, f** Morphology of single-cell clones of Canon neurons in A3 (top) and in A4 (bottom) obtained by MCFO. Dotted lines indicate segmental borders identified by the Fasciclin2 (Fas2)-positive TP1 tract[47]. **e** Dorsal view, **f** anterior view reconstructed from the region shown by a white box in **e**. The images are representative data from one of 12 independent experiments that gave similar results. **g** A focal plane showing that presynaptic sites of Canon neurons visualized by the expression of a presynaptic marker Syt-GFP (green) are also positive for ChAT (magenta, arrows). The image is a representative datum of one experiment. Scale bars, 50 μm (**e, f**) and 5 μm (**g**).

## Optogenetic activation of Canon neurons induces muscular relaxation.

Canon neurons are activated after MNs in each neuromere and send information via the ascending axons to anterior neuromeres, (i.e., in the opposite direction to backward propagation). Thus, Canon neurons may play roles in shutting down MN activities in anterior neuromeres during backward locomotion for instance by activating inhibitory premotor neurons. To study this, we conducted gain-of-function experiments via optogenetic activation of Canon neurons. Since *R91C05-Gal4* and *Canon-spGal4* drove expression not only in Canon neurons but also in other neurons, including Wave neurons, whose activation is known to induce backward locomotion and other escape

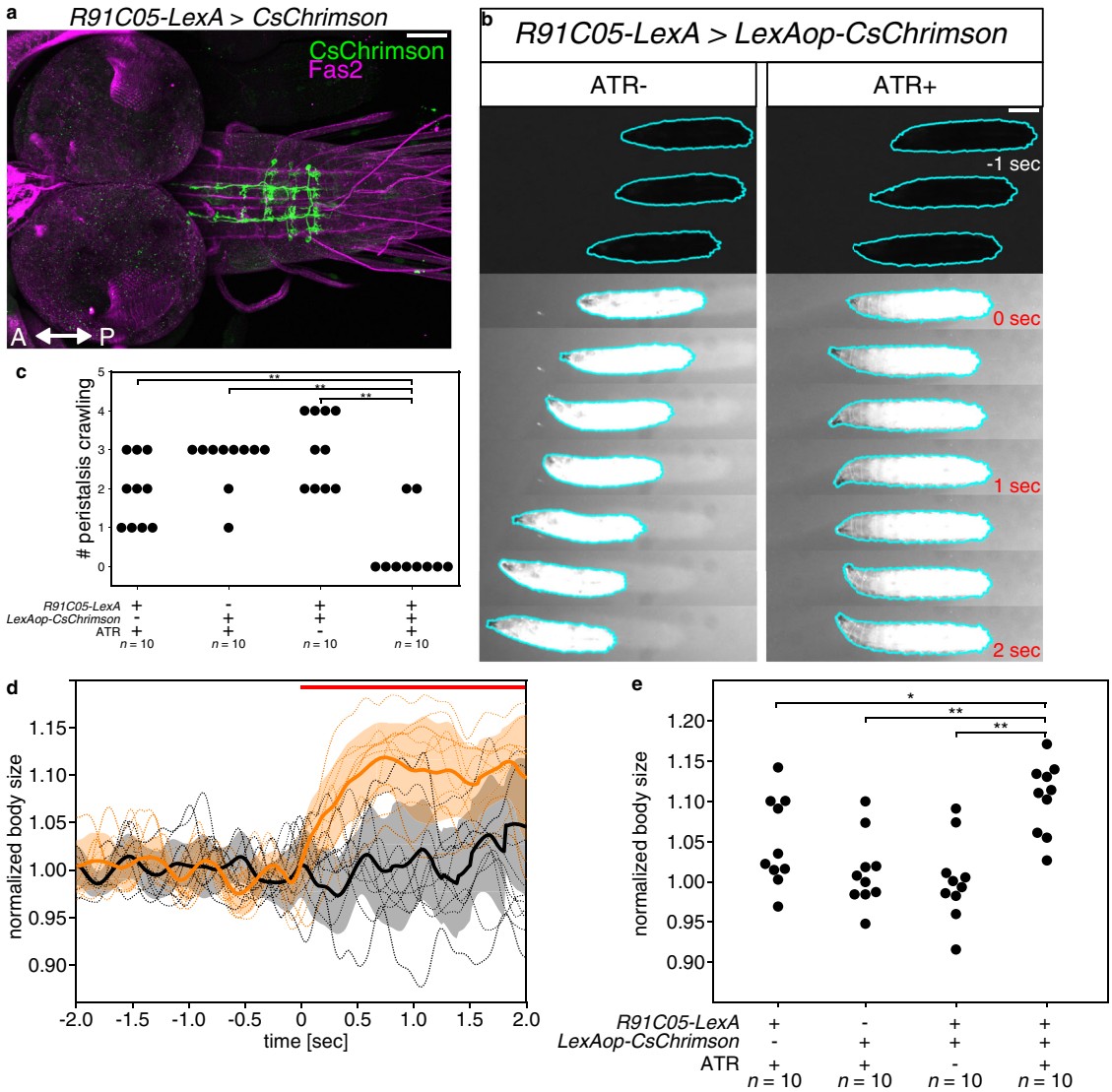

**Fig. 2 Activation of Canon neurons induces muscular relaxation. a** Expression of CsChrimson driven by *R91C05-LexA*. Note that the expression is only seen in Canon neurons. **b** Time-lapse images of control (left, ATR−) and experimental (right, ATR+) larvae during optogenetic perturbation. The experimental larva stopped immediately with its body relaxed when exposed to red light. 0 s indicates the onset of light stimulation. Cyan lines indicate contours of the larvae. **c** Quantification of the number of forward peristalses during the first 2 s of light stimulation. n = 10 larvae in each group. Eight out of ten larvae in the experimental group halted with their muscles relaxed upon photo-stimulation, while no control larvae stopped locomotion. **d** Change in the body size of the larvae after photostimulation (orange, experimental; black, control ATR−). The central thick lines represent mean values. The error bands (shaded area) show standard deviation. Dotted lines indicate values of individual larvae. Red bar shows the duration of photostimulation. **e** Quantification of the normalized size of the larvae during the first 2 s of light stimulation. n = 10 larvae in each group. *p < 0.05, **p < 0.01. Comparison with LexAop−, p = 4.29 × 10⁻³; comparison with LexA−, p = 4.29 × 10⁻³; comparison with ATR−, p = 4.29 × 10⁻³, the two-sided Fisher exact test with the Holm method (**c**). Comparison with LexAop−, p = 3.12 × 10⁻²; comparison with LexA−, p = 3.41 × 10⁻³; comparison with ATR−, p = 3.02 × 10⁻³, the two-sided Mann–Whitney U-test with the Holm method (**e**). Scale bars, 50 μm (**a**) and 1 mm (**b**).

behaviors[20], they were not suitable for the gain-of-function analyses (Supplementary Fig. 1a, c and Supplementary Fig. 2a, b). However, we found that *R91C05-LexA* drove expression only in Canon neurons in neuromeres A1–A3 (Fig. 2a), and used this for the gain-of-function analysis. We expressed CsChrimson, a red-shifted Channelrhodopsin[33], in Canon neurons, and optogenetically activated these neurons with short (5 s) light pulses. When photostimulation was applied to control larvae, all animals continued forward locomotion since they are insensitive to red light. In contrast, when light was applied to experimental larvae undergoing forward locomotion, a majority of them (8 out of 10) halted forward crawl within 2 s (Fig. 2b, c and Supplementary Movie 3). There were also changes in the overt body shape as the

larvae halted: the entire body became relaxed upon photo-stimulation and remained so during the duration of light application (Fig. 2b, d, e). We also studied the effects of optogenetic activation on body muscles in dissected larvae expressing *Mhc > GFP*[34] and found that muscles became relaxed upon photo-stimulation (Supplementary Fig. 3). Taken together, these results indicate that activation of Canon neurons induces relaxation of body muscles.

**Canon neurons are required for proper muscular relaxation.** We next studied whether Canon neurons are required for muscular relaxation during backward locomotion by conducting loss-

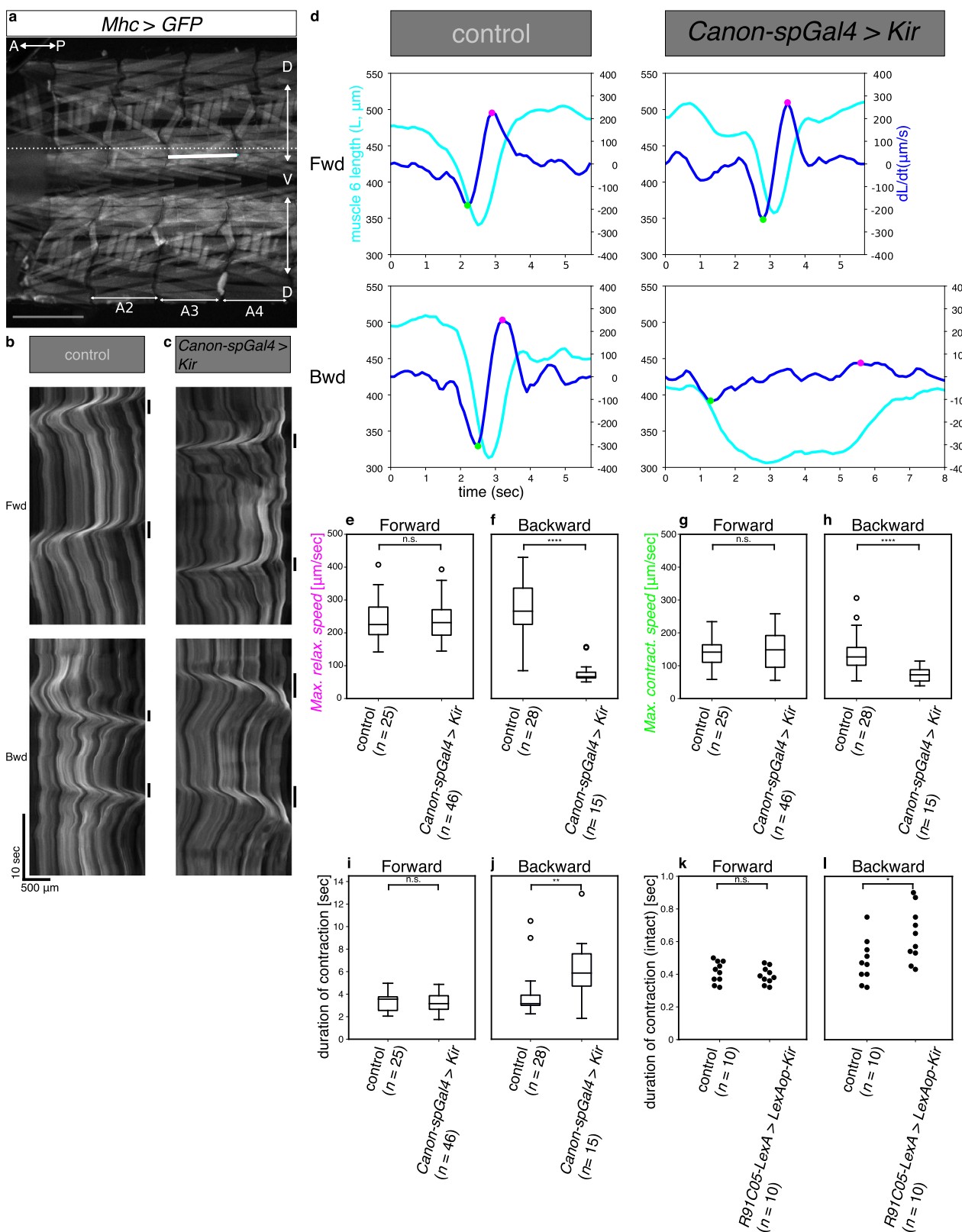

of-function experiments. We used *Canon-spGal4* to express Kir2.1 (inwardly-rectifying potassium channel)[35] in Canon and Wave neurons to suppress baseline excitability, and observed muscle dynamics in dissected larvae undergoing spontaneous forward and backward peristalses (Fig. 3a). We found that the muscular relaxation was disrupted and peristalses duration were prolonged during backward peristalses, but not forward

peristalses, when Canon neurons' activity was suppressed (Fig. 3b, c and Supplementary Movie 4).

To examine the phenotype more closely at the cellular level, we focused on a ventral longitudinal (VL) muscle (muscle 6) in segment A3 and measured the change in length of the muscle during peristalses (Fig. 3d). We then obtained the maximum speed of muscular relaxation and contraction and the duration of

**Fig. 3 Canon neurons are required for proper muscular relaxation during backward peristalses. a–j** Analyses of muscular relaxation in dissected larvae. **a** An image of body-wall muscles visualized by expression of *MhcGFP*, focusing on ventral muscles in segments A2–A3. The white solid line indicates the length of muscle 6 and the white dotted line indicates the location used to make the kymographs shown in **b**, **c**. The image is representative datum from one of six independent experiments that gave similar results. **b**, **c** Kymographs showing movement of the ventral muscles in control (Gal4 negative condition) (**b**) and *Canon-spGal4 > UAS-Kir* (**c**) larvae. Bars on the right side of each kymograph indicate the duration of backward and forward peristalses, respectively. Note elongated duration of backward peristalses in the experimental animal. The images are representative data of three experiments with control animals, and of three experiments with experimental animals, respectively. **d** Plots of muscular length (cyan line) and its time differential (blue line) during forward (top) and backward peristalses (bottom). Magenta and green points indicate the time points when relaxation (magenta) and contraction (green) speed is maximum, respectively. **e–h** Comparison of the maximum muscular relaxation (**e**, **f**) and contraction (**g**, **h**) speed during forward and backward peristalses. **i**, **j** Comparison of duration of muscular contraction during forward (**i**) and backward (**j**) peristalses. $n = 25$ forward and 28 backward peristalses in three control larvae and 46 forward and 15 backward peristalses in three experimental larvae. **k**, **l** Comparison of the duration of contraction in A2 segment during forward (**k**) and backward (**l**) locomotion of intact larvae. $n = 10$ forward and 10 backward locomotion from ten larvae in each group. *$p < 0.05$, **$p < 0.01$, ****$p < 0.0001$, n.s. not significant, $p > 0.05$. $p = 1.42 \times 10^{-7}$ (**f**), $p = 4.32 \times 10^{-5}$ (**h**), $p = 2.70 \times 10^{-3}$ (**j**), $p = 4.11 \times 10^{-2}$ (**l**), the two-sided Mann–Whitney $U$-test. Center line, median; box limits, upper and lower quartiles; whiskers, maximum and minimum between 1.5× interquartile range; points, outliers (**e–j**). Scale bars, 500 μm (**a**, **b**).

muscular contraction, and compared the values between forward and backward peristalses and between the control and experimental groups (Fig. 3e–j). The quantification revealed that maximum relaxation speed was dramatically decreased and the duration of muscular contraction was prolonged during backward, but not forward peristalses when Canon neuron function was inhibited. The result provides evidence that Canon neurons regulate relaxation of muscles during backward peristalses. There was also a slight decrease in contraction speed during backward peristalses when Canon neurons were silenced (Fig. 3h). Since Canon neurons are not active during the muscle contraction phase, this is likely an indirect consequence of the delayed muscular relaxation: it is well known that sensory feedback modulates the speed of locomotion, and it has been suggested that completion of muscular contraction and relaxation in one segment may activate the premotor circuits in the next segment via sensory feedback[34].

The above experiments were conducted using *Canon-spGal4*, which targets Wave neurons in addition to Canon neurons. To exclude the involvement of Wave neurons in the observed phenotype and to study the role of Canon neurons in muscular relaxation in intact larvae, we next drove expression of Kir2.1 specifically in Canon neurons by using *R91C05-LexA* and studied muscular relaxation during forward and backward peristalsis (backward peristalsis was induced by blue light stimulation). As in the dissected preparation, the duration of muscular contraction was increased in the experimental intact larvae during backward but not forward locomotion (Fig. 3k, l). The result indicates inhibition of Canon not Wave neurons is responsible for the observed phenotype. A similar increase in the duration of muscular contraction was also seen when tetanus toxin (TeTxLC)[36] was expressed in Canon neurons by *Canon-spGal4* to suppress synaptic transmission (Supplementary Fig. 2c, d). Involvement of Wave neurons was further excluded by the observation that inhibiting Wave activity by using an independent Wave-specific Gal4 driver (*MB120B-spGal4*) had no effect on the muscular contraction or relaxation during forward and backward peristalses (Supplementary Fig. 2e, f). Taken together, these results indicate that Canon neurons are required for proper muscular relaxation during backward peristalses.

**Canon neurons receive inputs from command systems for backward locomotion**. To further study how Canon neurons regulate muscular relaxation during backward locomotion, we elucidated the circuit structure around Canon neurons using reconstructions from a serial section transmission electron microscopy (ssTEM) image data set of an entire first instar larval CNS[37]. Since premotor circuits were the likely targets of Canon

neurons and previous studies comprehensively reconstructed 1st-order premotor interneurons in neuromere A1[27,38,39], we started reconstruction of Canon neurons in neuromeres A2 and A3, which arborize presynaptic sites in neuromere A1. We first uniquely identified Canon neurons in the existing ssTEM reconstruction based on the morphological features obtained from the single-cell MCFO clones (Fig. 1e, f), and found them annotated as "A18g" (Fig. 4a). We then extended the reconstruction to upstream and downstream synaptic partners of the Canon neurons (Figs. 4–6 and Supplementary Figs. 4–6, 8). The upstream neurons with the largest number of synaptic connections with the Canon neurons were: (1) thoracic descending neurons (ThDNs), (2) A27l interneurons, and (3) Canon neurons themselves (Fig. 4b–f and Supplementary Fig. 4a, b). ThDNs, descending neurons present in the thorax, are major components of the downstream circuits of MDNs, command neurons of backward locomotion located in the brain (Fig. 4f)[23]. A27l neurons, segmentally-repeated interneurons present in the VNC, are major downstream partners of ThDNs[23]. In addition, Canon neurons receive direct inputs from MDNs. Thus, Canon neurons receive large synaptic inputs directly or indirectly from MDNs (Fig. 4f and Supplementary Fig. 4b). These results suggest that Canon neurons are recruited during backward locomotion by the backward command system. In addition to the MDN command system, Canon neurons receive major inputs from Canon neurons in other neuromeres, and thus form a bidirectionally connected circuit composed of segmental homologs (hereafter called Canon–Canon network) as will be detailed below. Thus, two major sources of inputs to Canon neurons are the MDN command system and the Canon–Canon network (Fig. 4f).

**Canon neurons provide outputs to inhibitory premotor circuits**. Major downstream partners of the Canon neurons include: (1) first-order premotor interneurons A06a, A06c, A31d, and A31k, (2) second-order premotor interneurons A02l and GVLIs, (3) Canons themselves, and (4) intersegmental neurons located in the thorax or subesophageal zone (SEZ) named post Canon neuron 04, 05, and 23 (Fig. 5a–e, Supplementary Fig. 4c, d, and Supplementary Fig. 5a–h). Consistent with the notion that Canon neurons regulate muscular relaxation, all of the four first-order premotor interneurons appear to be inhibitory. A06a, A06c, and A31k were previously identified as putatively inhibitory since they express GABA[38,39]. Another downstream premotor neuron A31d was putatively GABAergic because it belongs to the same lineage as A31b and A31k—premotor interneurons previously identified as GABAergic—and shares morphological characteristics to these neurons[26,38,39]. Functional analyses showed that at least one of the four classes of premotor interneurons, A31k, indeed inhibit

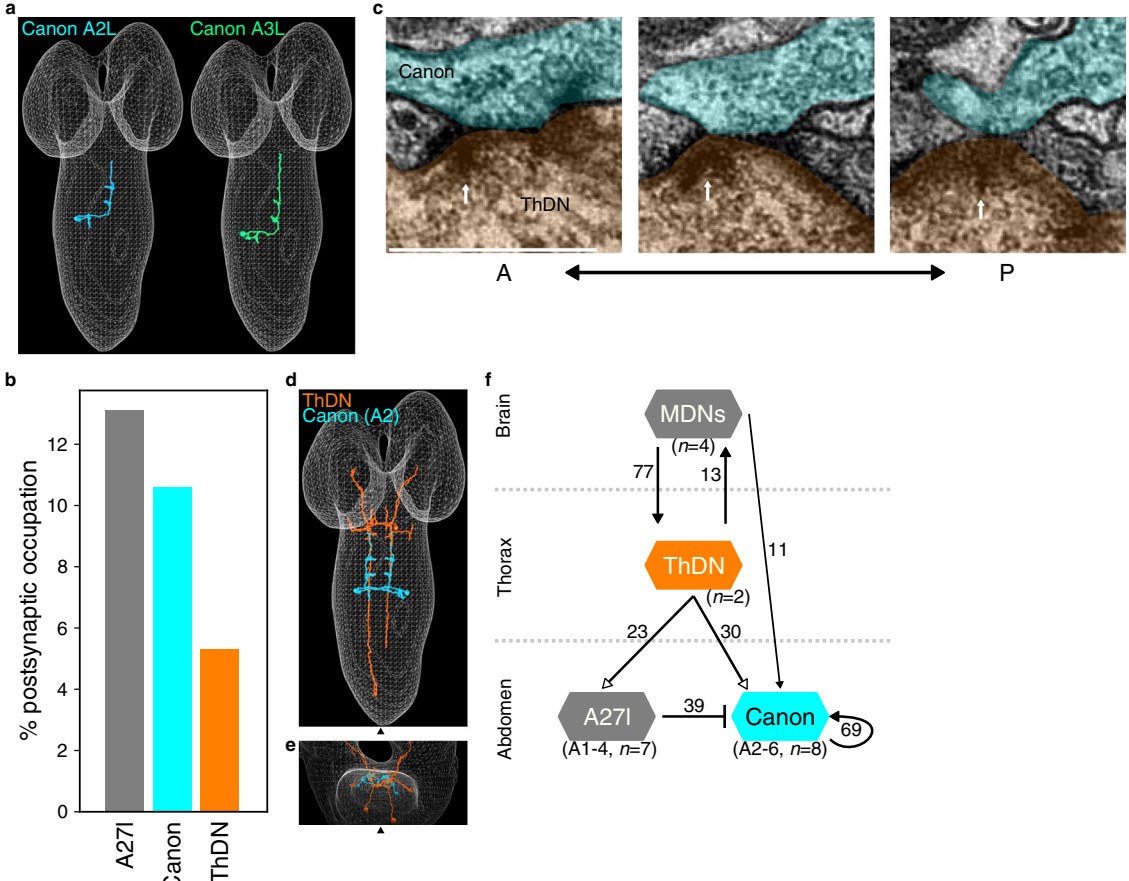

**Fig. 4 Upstream circuits of Canon neurons. a** EM reconstruction of left side Canon neurons in neuromeres A2 and A3. **b** A bar graph showing percent of total postsynapses of A2–A3 Canon neurons that are targeted by the indicated upstream neuron. Major partners (occupying more than 5% of total postsynapses) are shown. **c** Serial EM images showing synapses between ThDN neuron (presynapse, orange) and A2 Canon neuron (postsynapse, light blue). White arrows indicate T-bars. The images show representative synapses of observed 30 synapses from ThDN to Canon neurons. **d**, **e** Dorsal (**d**) and anterior view (**e**) of reconstruction images of ThDNs (orange) and A2 Canon (blue) neurons. Triangles indicate the midline. **f** A diagram of connectivity from MDN to Canon neurons. Arrows with filled heads indicate cholinergic synapses while arrows with empty heads indicate synapses with unidentified neurotransmitter. The bar indicates inhibitory outputs. Numbers next to the lines show the number of synapses. Scale bar (**c**), 500 nm.

motor activity. Optogenetic activation of A31k neurons in behaving larvae induced rapid relaxation of the body as observed when Canon neurons are activated (same results were obtained when two independent sparse Gal4-drivers targeting A31k were used; Fig. 5f–h and Supplementary Fig. 5i, j). Each of the four classes of premotor interneurons in A1 received inputs from two Canon neurons located in neuromeres A2 and A3 and send outputs to MNs, most of which innervate longitudinal muscles located in the segments A1 and A2 (Fig. 5e). This circuit configuration predicts Canon neurons provide late-phase and prolonged inhibition to MNs during backward locomotion (see "Discussion" section).

The second-order premotor neurons A02l and GVLI are segmentally repeated interneurons (Supplementary Fig. 5a, b)[17,38,40,41]. GVLIs not only receive inputs from Canon neurons directly but also indirectly via A02l neurons, suggesting that these two second-order premotor neurons function in closely related circuits (Fig. 5e). A previous study showed GVLI neurons are glutamatergic neurons whose optogenetic activation elicits muscular relaxation[40]. While GVLIs formed direct synaptic contacts with MNs, a majority of the downstream targets were first-order premotor interneurons including the premotor interneuron, A31d, which is also a direct target of Canon neurons (Fig. 5e). GVLIs show activities during fictive forward[40] and backward (this study, Fig. 5i) locomotion. To summarize, major

downstream targets of Canon neurons are Canon–Canon network and inhibitory premotor circuits including the first order and second order premotor interneurons, through which Canon neurons may induce muscular relaxation (Fig. 5e).

**Canon–Canon network regulates its own intersegmentally propagating activities.** As mentioned above, Canon neurons connect bidirectionally with other Canon neurons (Fig. 6a–e and Supplementary Fig. 4b, d). We first noticed connections between Canon neurons in segments A2 and A3: the right Canon neuron in A2 forms synapses with the right Canon in A3 bidirectionally, in the bundle consisting of Canon neurons' ascending axons. We extended the reconstruction to Canon neurons in neuromeres A4 and A5 and found bidirectional synaptic connections between all combinations of the four Canon neurons in A2–A5 (Supplementary Fig. 6). No synapses were seen between the right and left Canon neurons.

To study the roles of the Canon–Canon network in generating the wave-like activity of Canon neurons, we conducted loss-of-function analyses of the outputs of Canon neurons. In this experiment, we inhibited Canon neurons' chemical transmission by expressing TeTxLC and observed the Canon neurons' activity using calcium imaging. In this condition, inputs from other upstream neurons to the Canon circuits are not perturbed (Fig. 6e, f). If the pattern of Canon neurons' activity is solely

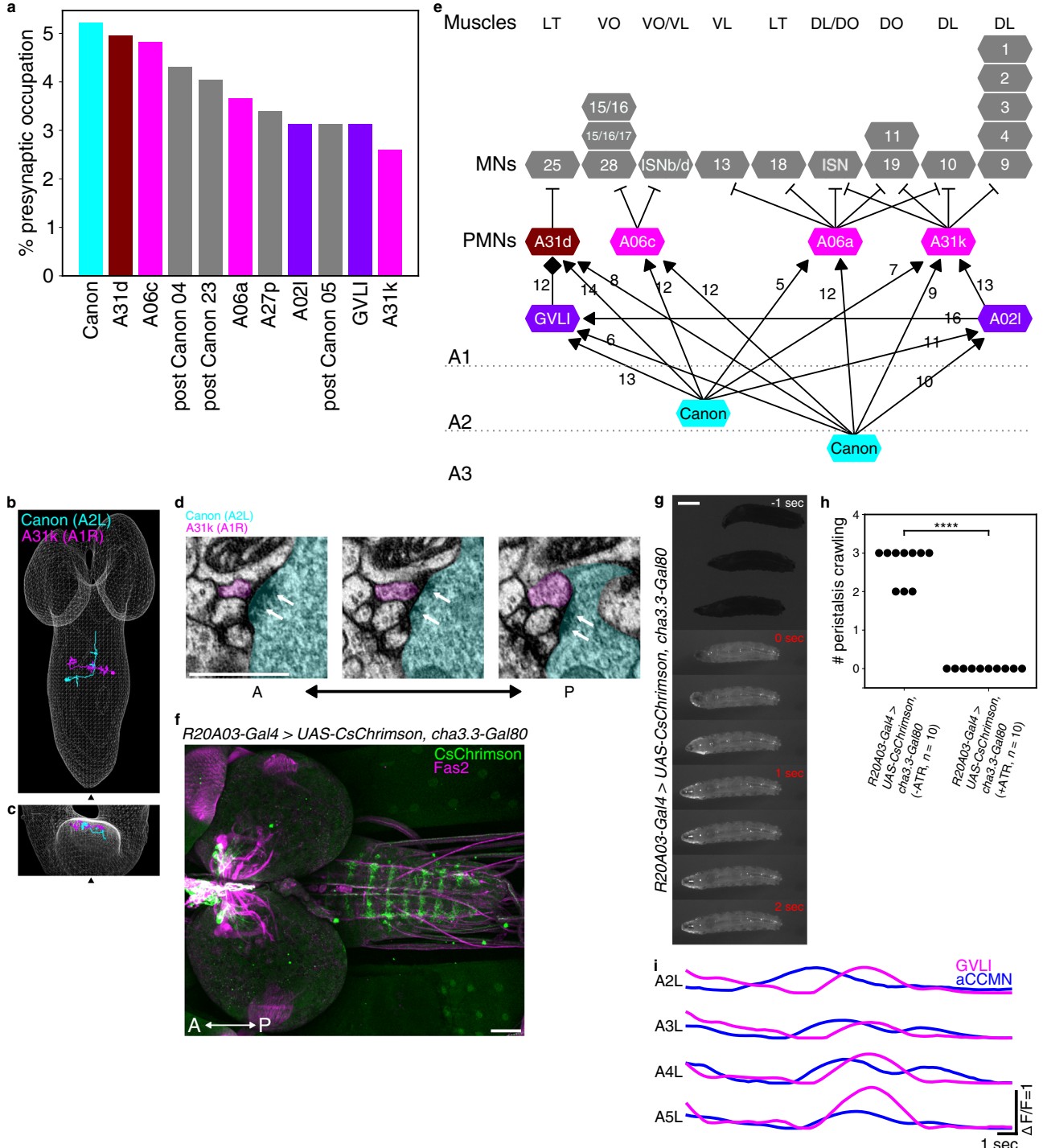

**Fig. 5 Downstream circuits of Canon neurons. a** A bar graph showing percent of total presynapses of A2–A3 Canon neurons that target the indicated downstream neuron. Major partners (occupying more than 2.5% of total presynapses) are shown. **b, c** Dorsal view (**b**) and anterior view (**c**) of EM reconstructed A31k and Canon neurons. **d** Serial EM images showing synapses between Canon (cyan) and A31k (magenta) neurons. White arrows indicate T-bars. The images show representative synapses of observed 20 synapses from Canon to A31k neurons. **e** A circuit diagram from Canon neurons to muscles. Gray dotted lines indicate boundaries of neuromeres. Arrows with triangle heads indicate excitatory synaptic connections, lines with bar heads inhibitory synaptic connections and those with diamond-shaped heads glutamatergic synapses. Numbers next to the lines show the number of synapses. **f** Expression of CsChrimson driven by *R20A03-Gal4, cha3.3-Gal80*. The image shows representative datum from one of three independent experiments that gave similar results. **g** Time-lapse images of experimental larvae during optogenetic activation. The larva halted immediately with its body relaxed when exposed to red light. 0 s represents the onset of photostimulation. **h** Quantification of the number of forward peristalses during the first 2 s of light stimulation. $n = 10$ larvae in each group. All larvae in the experimental group stalled upon light stimulation, while control (ATR−) larvae were unaffected. ****$p < 0.0001$. $p = 1.08 \times 10^{-5}$, the two-sided Fisher exact test. **i** Plots of calcium signals in aCC MNs and GVLIs during fictive backward locomotion. Scale bars, 500 nm (**d**), 50 µm (**f**), and 1 mm (**g**).

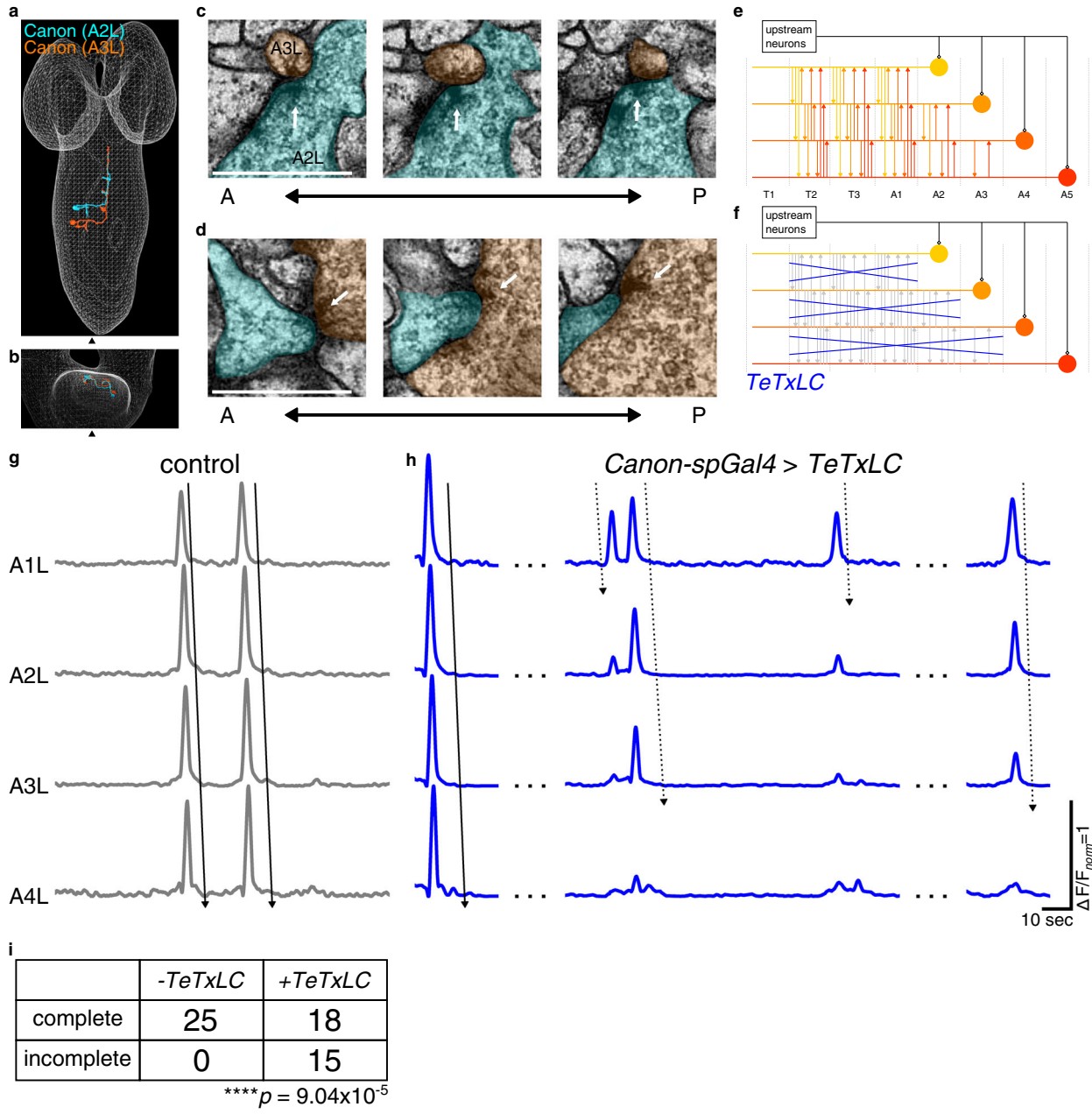

**Fig. 6 Canon–Canon network regulates its own wave-like activity. a, b** EM reconstructed Canon neurons in A2 and A3. Note that these neurons form a bundle of ascending axons. Dorsal (**a**) and anterior (**b**) view. **c, d** Serial EM images showing bidirectional synapses between Canon neurons (Cyan: A2, Orange: A3). White arrows indicate T-bars. The images show representative synapses of observed five synapses from A2 to A3 Canon neuron and of observed 13 synapses from A3 to A2 Canon neuron. **e, f** Schemes of Canon–Canon networks in control (**e**) and *Canon > TeTxLC* (**f**) larvae. Canon neurons connect with each other via synapses present in their ascending axons. TeTxLC disrupts Canon–Canon transmission but leaves input from other upstream neurons intact. **g, h** Activity propagation among Canon neurons in control (**g**) and *Canon > TeTxLC* (**h**) larvae. Arrows with solid lines indicate complete waves, while those with dotted lines indicate incomplete waves. **i** A cross table showing the number of complete and incomplete waves in the control (*−TeTxLC*) and *Canon > TeTxLC* (*+TeTxLC*) larvae. The number of incomplete waves is significantly increased in the experimental group. $n = 25$ waves in four control larvae and 33 waves in seven experimental larvae. ****$p < 0.0001$. $p = 9.04 \times 10^{-5}$, the one-sided chi-squared test. Scale bars, 500 nm (**c, d**).

generated by upstream neurons other than Canon neurons, the Canon neurons should show wave-like activity even if Canon neuron synapses are suppressed. However, if synaptic transmission between Canon neurons is required for the wave-like activity, it should be disrupted in this condition. We found that the latter was the case (Fig. 6g–i). When synaptic transmission by Canon neurons was inhibited, wave-like activity in Canon neurons was initiated but often failed to propagate all the way to the posterior end. Again, the experiment was conducted using

*Canon-spGal4*, which targets Wave neurons in addition to Canon neurons, not the Canon-specific *R91C05-LexA*, since expression driven by *R91C05-LexA* was not strong enough for this experiment. However, Wave neurons are unlikely to be involved in the regulation of Canon activity, since they are not active during spontaneously occurring fictive locomotion (Supplementary Fig. 7a)[20]. These results suggest that synaptic transmission between Canon neurons is necessary for their own wave-like activity. However, we do not exclude the possibility that other

downstream targets of Canon neurons provide feedback to Canon neurons in other segments, for instance via proprioception or gap junctions.

It is important to note that propagation of excitatory drive was not terminated before reaching the posterior end in the experimental animals (*Canon > TeTxLC*). As in dissected *Canon > Kir2.1* larvae (Fig. 3b, c), the wave of muscular contraction was slower due to prolonged muscle contraction but propagated to the posterior ends (Supplementary Fig. 7b–d). Thus, inhibition of synaptic transmission by Canon neurons specifically inhibits the propagation of their own activity crucial for muscular relaxation, but leaves the propagation of the excitatory drive intact.

## Discussion

Our study identified a class of premotor interneurons, the Canon neurons, that regulate muscular relaxation, and revealed their cellular-level circuit structure, including the upstream backward command neurons and downstream inhibitory premotor circuits extending from the brain to the muscles. Moreover, we found that Canon neurons form synapses with each other to constitute a self-regulating circuit. Our results suggest that the Canon–Canon network is important, not only as an actuator of muscular relaxation in each segment, but also as the pattern regulator of its own propagating activity during backward locomotion.

**Regulation of premotor inhibition and muscular relaxation.** For animal movements to occur in a patterned and coordinate manner, timing of not only muscle contraction, but also muscle relaxation must be precisely controlled. In locusts, inhibitory GABAergic motor neurons have been shown to regulate muscle performance to optimize behavioral performance in a variety of contexts including postural control, predatory strikes, and escape movements[12,42]. Similarly, in the nematode *C. elegans*, phase-shifted oscillatory calcium activities in excitatory and inhibitory motor neurons regulate muscle contraction and relaxation[13,43]. Motor neuron activity is also dependent on the activity of excitatory and inhibitory interneuron populations. While previous studies have identified a number of interneurons and circuits that regulate muscle contraction in various model systems, neural circuit mechanisms underlying the timing of muscle relaxation are less well explored[6–8]. Here, we revealed crucial roles played by Canon neurons in the regulation of muscle relaxation during backward locomotion of *Drosophila* larvae. The following evidence points to the role of Canon neurons in muscle relaxation, not contraction. First, Canon neurons are activated much later than the downstream MNs. Calcium imaging showed that Canon neurons are active later than MNs in the same neuromere and at a similar timing as MNs in the next or third neuromere during fictive backward locomotion (corresponding to a phase delay of 1–2 neuromeres; Fig. 1d and Supplementary Fig. 1e). Additionally, EM reconstruction shows that Canon neurons (via premotor interneurons) connect to MNs in the next or third neuromere, in the opposite direction to backward wave propagation (generating a delay of 1–2 neuromeres; Fig. 5). Taken together, Canon neurons are activated with a delay corresponding to propagation in 2–4 neuromeres to their target MNs during fictive locomotion. This late onset of Canon activities implies roles in muscle relaxation not excitation. Second, optogenetic activation of Canon neurons induces muscle relaxation in the larvae (Fig. 2b–e and Supplementary Fig. 3). Third, loss-of-function of Canon neurons impaired muscle relaxation during backward peristalses of dissected and intact larvae (Fig. 3 and Supplementary Fig. 2c, d). This indicates that orchestrating muscle relaxation pattern during backward waves is an active process requiring the activity of Canon neurons. Finally, Canon neurons target a number of

premotor interneurons, which can potentially inhibit MNs. Since Canon neurons are cholinergic and thus likely excitatory, their activation, and in turn, downstream inhibitory premotor interneurons, can terminate MN activity and thus induce muscle relaxation (Fig. 5). The fact that activation of GVLIs or A31k, like that of Canon, induces muscular relaxation is consistent with this idea. Taken together, our results reveal a circuit motif regulating muscle relaxation consisting of a second-order excitatory premotor interneuron (Canon) and a group of downstream inhibitory premotor interneurons, including both first-order (such as A31k) and second-order (such as GVLI) premotor neurons. While our results are consistent with this model for Canon neuron function on the time scales of activation studied, it is important to note that Canon neurons also connect to a large number of other interneurons that are not premotor inhibitory cells. This suggests that Canon neurons may also have additional functional roles mediated by other downstream partners.

Larval musculature consists of two groups of antagonistic muscles: longitudinal muscles that contract the body along the anterior–posterior axis, and transverse muscles that contract the body circumferentially[44]. These two groups of antagonistic muscles are activated at different times with the longitudinal muscles contracting earlier[21,45] during peristalses. Premotor interneurons immediately downstream of Canon neurons described above provide inhibitory inputs largely to longitudinal muscles rather than transverse muscles, suggesting that Canon neurons may primarily regulate the relaxation of a specific functional type of muscle within the neuromuscular system (Fig. 5e).

**Circuits mediating early-phase and late-phase motor inhibition.** As mentioned in the Introduction, previous studies identified Ifb-Fwd and Ifb-Bwd neurons, which like Canon neurons target a group of inhibitory premotor neurons during forward and backward locomotion, respectively[25]. Ifb-Fwd and Ifb-Bwd neurons are active at a similar timing to MNs in the same neuromere and provide excitation to PMSIs, and other shared downstream premotor interneurons in the adjacent neuromere behind the motor propagation, which in turn provide inhibitory inputs to MNs innervating longitudinal muscles (L-MNs) and excitatory inputs to MNs innervating the antagonistic and later-activating transverse muscles (T-MNs, Fig. 7 and Supplementary Fig. 8a, b). Thus, during backward locomotion, timing of the activation of inhibitory premotor interneurons (and ensuing muscular relaxation) appears to be regulated via at least two pathways, one including Ifb-Bwd neurons and providing activation at a relatively early phase (~1 neuromere behind the wave front) and the other including Canon neurons and providing activation at a late phase (2–4 neuromeres behind the wave front; Fig. 7). It would be interesting to study in the future whether there exists a counterpart of Canon neurons during forward locomotion that mediates a late-phase inhibition.

Interestingly, Ifb-Bwd and Canon neurons provide inputs to almost non-overlapping sets of inhibitory interneurons, suggesting that dedicated premotor interneurons mediate early-phase versus late-phase motor inhibition (Supplementary Fig. 8c). A31k is the only downstream inhibitory premotor neurons common to Canon and Ifb-Bwd neurons. Activity of the A31k neuron lasts much longer than that of A02e (a PMSI), which is only innervated by Ifb-Bwd neurons, consistent with the idea that Ifb-Bwd and Canon neurons provides early-excitation and late-excitation to this and other downstream neurons, respectively (Supplementary Fig. 8d–h).

Inhibition mediated by Canon neurons is not only later but also likely longer compared to that mediated by Ifb-Bwd neurons,

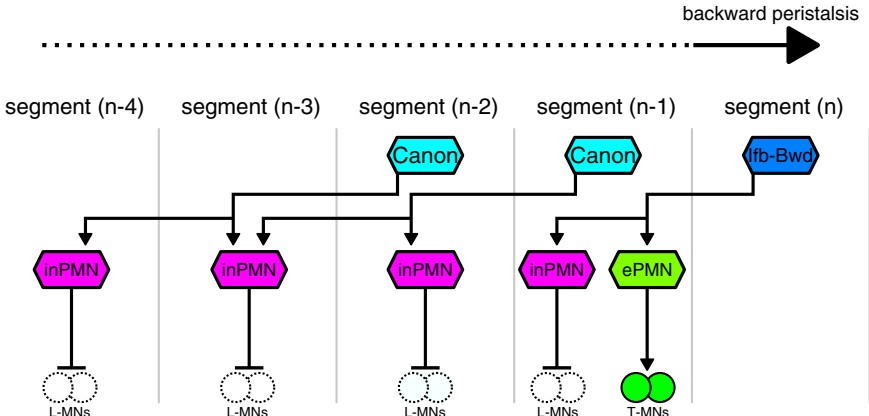

**Fig. 7 A model of early- and late-phase motor inhibition mediated by Ifb-B and Canon neurons during backward locomotion.** When the wave front reaches the segment (*n*), Ifb-Bwd neurons in segment (*n*) and Canon neurons in segment (*n*−1) and segment (*n*−2) are activated. Ifb-Bwd and Canon neurons in turn activate inhibitory premotor neurons (inPMN) innervating L-MNs in segment (*n*−1) and in segments (*n*−2) to (*n*−4), respectively, via their ascending axons. Ifb-Bwd neuron activates excitatory premotor neurons (ePMN) that innervate T-MNs. Neurons outlined by dotted or solid lines indicate inactive or active neurons, respectively. Arrows with filled heads indicate excitatory synapses while those with bar heads indicate inhibitory outputs.

since the target premotor interneurons in each neuromere receive inputs from Canon neurons in two consecutive neuromeres. The long duration of Canon-mediated activation suggests that it is present even after MN activity ceases. Thus, Canon-mediated late-phase inhibition may play roles not only in the process of muscular relaxation but also maintenance of the relaxed state after movement, which might be important to inhibit inappropriate contraction induced by self-generated sensory signals[46].

**Canon–Canon network regulates its own intersegmentally propagating activities**. Our results showed that Canon neurons regulate not only muscle relaxation but also their own intersegmentally propagating activities. Segmentally repeated Canon neurons bidirectionally connect with each other by forming a bundle of axons (Fig. 6a–e). When synaptic transmission in the Canon–Canon network was inhibited, segmentally propagating activity of Canon neurons was often interrupted (Fig. 6f–i). It should be noted that in the absence of transmission by the Canon–Canon network, waves of muscle contraction propagated normally in all peristalses we observed (Supplementary Fig. 7b–d). Thus, the Canon–Canon network is specifically required for generating the pattern of muscular relaxation, not contraction. However, the activity of Canon neurons should also be under the influence of the earlier activating excitatory CPGs, since Canon activities and muscle relaxation must occur in phase with contraction of muscles. Indeed, calcium imaging showed that propagating activity of Canon neurons is in phase with that of earlier activating MNs (Fig. 1c, d and Supplementary Fig. 1e). Thus, the Canon–Canon network appears to function as a sub-circuit or smaller pattern generating circuit that functions within a larger CPG, as proposed in other systems[1,3].

Canon neurons are activated later than MNs in the same neuromere with a lag of one to two neuromeres (Fig. 1c, d and Supplementary Fig. 1e). In contrast to Canon neurons, the activities of Ifb-Fwd and Ifb-Bwd neurons, which also mediate motor inhibition, occur at a similar timing as MNs in the same neuromere and are likely under direct influence of the CPGs driving motor excitation. Delay appropriate for motor inhibition is realized in the Ifb-Fwd and Ifb-Bwd pathway by projecting an intersegmental axon in an opposite direction to the motor wave. Similarly, Canon neurons send intersegmental axons to connect with premotor neurons in neuromeres behind the motor wave, so as to activate the downstream neurons with a delay corresponding to one or two neuromeres compared to the wave front.

However, to generate even longer delays in muscle relaxation, Canon neurons may have evolved to form a pattern generating circuit, which is capable of generating delayed propagating activities distinct from the excitatory activity propagation.

How the Canon–Canon network regulates propagating waves of activity remains to be determined. We found that Canon neurons receive strong inputs from ThDNs neurons which in turn are activated by MDNs, command-like neurons for backward locomotion. The combination of these descending inputs, Canon–Canon signaling, and some other signals from the excitatory CPGs may generate the delayed propagating activity for muscle relaxation. Further exploration of the connectome and circuit mechanisms underlying the delayed propagation will be important future goals. It will also be interesting to study if similar circuit motifs as the Canon–Canon network regulate the timing of muscle relaxation in other invertebrate and vertebrate systems. Prolonged muscle contraction and delayed relaxation, as observed in larval preparations with inhibited Canon function, is a common symptom found in various human movement disorders[8–10]. Further investigation of Canon–Canon network and related circuits in other species may improve our understanding of the control of muscle relaxation in health and disease.

## Methods

***Drosophila melanogaster* strains**. Third-instar larvae of the following fly strains were used for all functional and histological experiments. *y¹ w¹¹¹⁸* (Bloomington *Drosophila* Stock Center (BDSC), #6598), *R91C05-Gal4* (BDSC, #40578), *R26A08-Gal4*[40] (BDSC, #49153), *R20A03-Gal4* (BDSC, #48871), *RRa-Gal4*[48] (gift from Dr. Miki Fujioka), *UAS-GCaMP6s*[49] (BDSC, #42746), *UAS-CD4::GCaMP6f*[25], *UAS-TeTxLC*[36] (BDSC, #28838), *UAS-Kir2.1::EGFP*[35] (BDSC, #6596), *UAS-CD4::tdGFP*[50] (BDSC, #35836), *UAS-DenMark, UAS-syt::GFP/CyO; D/TM6C* (*UAS-TLN-21*)[31] (gift from Dr. Bassem A. Hassan), *MCFO4*[30] (BDSC, #64087), *R91C05-LexA* (BDSC, #61629), *LexAop-CsChrimson::mVenus*[33], (BDSC, #55136), *UAS-VNC-CsChrimson* (*20xUAS>dsFRT>CsChrimson::mVenus in attP18; tsh-LexA, pJFRC79-8xLexAop2-FlpL in attP40*, this study), *MhcGFP*[34] (gift from Dr. Cynthia L. Hughes), *Canon-spGal4* (A combination of *R91C05-Gal4.AD* (BDSC, #70979) and *VT019059-Gal4.DBD* (BDSC, #71731)), *MB120B-spGal4*[20], *Cha3.3kbp-Gal80*[51] (gift from Dr. Toshihiro Kitamoto).

*R91C05-Gal4* targets Canon neurons in abdominal neuromeres A3–A5, wave neurons, and some neurons whose cell bodies are located on the lateral side of the ventral nerve cord. *Canon-spGAL4* targets Canon neurons in abdominal neuromeres A1–A6, Wave neurons, and putative type II neuroblasts in the brain[52]. *R91C05-LexA* targets Canon neurons in abdominal neuromeres A1–A4, but driven expression could be weak or absent in some of these neuromeres.

*R20A03-Gal4, cha3.3-Gal80* targets only A31k neurons in the VNC and a few other neurons in the brain. In the progeny of *R20A03-Gal4* crossed with *UAS-VNC-CsChrimson*, CsChrimson was expressed in A31k neurons and a few other neurons in the VNC but no neurons in the brain.

**Calcium imaging**. The larvae were dissected and their CNSs were isolated and placed on a MAS-coated slide glass (S9215, Matsunami Glass, Japan) in TES buffer (TES 5 mmol, NaCl 135 mmol, KCl 5 mmol, CaCl$_2$ 2 mmol, MgCl$_2$ 4 mmol, Sucrose 36 mmol). The imaging was conducted with an EMCCD camera (iXon, ANDOR TECHNOLOGY, UK) mounted on an upright microscope (Axioskop2 FS, Zeiss, Germany) equipped with a spinning-disk confocal unit (CSU21, Yoko-gawa, Japan), and a 20× water immersion objective lens (UMPlanFLN 20XW, Olympus, Japan) under the control of software (iQ 3.0, ANDOR TECHNOLOGY, UK). In experiments with GCaMP6s or GCaMP6f, 488 nm blue light (CSU-LS2WF, Solution Systems, Japan) with an intensity of 4.5 μW/mm$^2$ was used. The light intensity was measured by the laser power meter (LP1, Sanwa Electric Instrument, Japan). Exposure time and frequency (frames per second, fps) were 99.1 ms and 10 fps for imaging of Canon neurons (Figs. 1b, 6g, h, and Supplementary Fig. 7a), 50 ms and 6 fps for simultaneous imaging of aCC MNs and Canon neurons (Fig. 1c, d, and Supplementary Fig. 1e), and 85 ms and 5fps for simultaneous imaging of aCC MNs and GVLIs (Fig. 5i). To rapidly acquire images from two focal planes each containing aCC MNs and Canon neurons or aCC MNS and GVLIs, a piezo unit (P-725, Physik Instrumente, Germany) controlled by a piezo Z controller (E-665 Piezo Amplifier/Servo Controller, Physik Instrumente, Germany) and a precise control unit (ER-PCUA-100, Andor Technology, UK) was used.

The regions of interest (ROIs) were set to surround the axons or cell bodies of the target neurons. Signal intensity in each frame was defined as the maximum or mean signal value among all pixels within each ROI. After extraction of raw signal data, a Savitzky–Golay filter was applied to the raw data to remove noise and perform smoothing. The calcium signals were normalized by the baseline intensity as follows,

$$\frac{\triangle F}{F} = \frac{F_{\text{ROI}[n]}(t) - Fb_{\text{ROI}[n]}(t)}{Fb_{\text{ROI}[n]}(t)},$$

where $n$ designates the ROI identity, $t$ the frame number, $F_{\text{ROI}[n]}(t)$ the signal intensity at $t$, and $Fb_{\text{ROI}[n]}(t)$ baseline intensity defined as the minimum $F_{\text{ROI}[n]}(t)$ between $t$ to $t + 100$.

**Analysis of phase lag during fictive locomotion**. To evaluate the time lag between the activity of aCC MNs and Canon neurons (Fig. 1d and Supplementary Fig. 1e), we focused on the initiation and peak of the rise in calcium signals in each fictive backward locomotion. The initiation was defined as the time when a rise in fluorescent started before the peak. For the analysis of activity timing of Canon and aCC MNs (Fig. 1d), peak time for neuron X (X = {aCCA2, aCCA3, aCCA4, aCCA5, aCCA6, CanonA3, CanonA4, CanonA5}),$t_{P_X}$,was normalized to the duration of wave propagation from A2 to A6 aCC MNs to give $t_{P_{Xnorm}}$ as follows.

$$t_{P_{Xnorm}} = \frac{t_{P_X} - t_{P_{aCCA2}}}{t_{P_{aCCA6}} - t_{P_{aCCA2}}}$$

For the analysis of activity duration of Canon and aCC neurons (Supplementary Fig. 1e), initiation time ($t_{I_X}$) for neuron X was also normalized to give $t_{I_{Xnorm}}$ as follow.

$$t_{I_{Xnorm}} = \frac{t_{I_X} - t_{I_{aCCA2}}}{t_{P_{aCCA6}} - t_{I_{aCCA2}}}$$

For the analysis of activity duration of A31k and A02e (Supplementary Fig. 8h), we used the data obtained in a previous research[25], in which GCaMP6f signals of A31k and A02e were recorded simultaneously with RGECO1 signals of A02e as a reference. Here normalized activity duration of A02e or A31k recorded with GCaMP6f located in A3 neuromere, $\tau_X$, is defined by the following equation.

$$\tau_X = \frac{t_{P_X} - t_{I_X}}{t_{P_{A02eRGECO1}} - t_{I_{A02eRGECO1}}}$$

**Immunohistochemistry**. The larvae were pinned on a silicon-filled dish and dissected in TES buffered saline. After dissection, the larvae were rinsed with phosphate buffered saline (PBS) and fixed in 4% paraformaldehyde in PBS for 30 min at room temperature. After a 30 min rinse with 0.2% Triton X-100 in PBS (PBT), the larvae were incubated with 5% normal goat serum (NGS) in PBT for 30 min. Then the larvae were incubated for at least 18 h at 4 °C with the primary antibodies mixed in with 5% NGS in PBT. After a 30 min rinse, the larvae were incubated at 4 °C with the secondary antibodies mixed in with 5% NGS in PBT for at least 12 h. Fluorescent images were scanned and acquired using a confocal microscope (FV1000, Olympus, Japan) with a 20× water immersion objective lens (UMPlanFLN 20XW, Olympus, Japan) or a 100× water immersion objective lens (LUMPlanFl 100XW, Olympus, Japan) under the control of software (Fluoview, Olympus, Japan). Antibodies:rabbit anti-GFP (Af2020, Frontier Institute; 1:1000; RRID: AB 2571573), guinea pig anti-GFP (Af1180, Frontier Institute; 1:1000; RRID:AB 2571575), mouse anti-FasII (1D4, Hybridoma Bank (University of Iowa); 1:10; RRID: AB 528235), rabbit anti-HA (C29F4, Cell Signaling Technology; 1:300; RRID: AB 1549585), rat anti-FLAG (NBP1-06712, Novus Biologicals; 1:200; RRID: AB 1625981), mouse anti-ChAT (4B1, Hybridoma Bank (University of Iowa); 1:50; RRID: AB 528122), rabbit anti-vGluT (Gift from Dr. Hermann Aberle; 1:1000;

RRID: AB 2315544), mouse anti-Brunchpilot (Brp) (nc82, Hybridoma Bank (University of Iowa); 1:50; RRID: AB 2314866), rabbit anti-DsRed (632496, Clontech; 1:500; RRID: AB 0015246), goat Alexa Fluor 488 anti-rabbit (A11034, Thermo Fisher Scientific; 1:300; RRID AB 2576217), goat Cy3 anti-rabbit (A10520, Thermo Fisher Scientific; 1:300; RRID:AB 10563288), goat Cy5 anti-rabbit (A10523, Thermo Fisher Scientific; 1:300; RRID: AB 2534032), goat Alexa Fluor 555 anti-mouse (A21424, Thermo Fisher Scientific; 1:300; RRID: AB 141780), goat Cy5 anti-mouse (A10524, Thermo Fisher Scientific; 1:300; RRID: AB 2534033), goat Alexa Fluor 633 anti-rat (A21094, Thermo Fisher Scientific; 1:300; RRID: AB 141553), goat Alexa Fluor 488 anti-guinea pig (A11073, Thermo Fisher Scientific; 1:300; RRID: AB 2534117), goat Alexa Fluor 647 anti-Horseradish Peroxidase (HRP) (123-605-021, Jackson ImmunoResearch; 1:200; RRID: AB 2338967).

**Optogenetics in behaving larvae**. In this study, CsChrimson[33] was used for optogenetical activation of neurons. The larvae were grown at 25 °C in vials with food, either containing 1 mmol of all-trans retinal (ATR) or none. The vials were covered with aluminum foil. Third instar larvae were gathered and washed to remove extraneous matter. Each behavior assay was conducted on an agar plate under shaded conditions. The surface temperature was adjusted at 25 °C ± 1 °C by a heating plate and the room temperature was also set at 25 °C ± 1 °C. For optogenetic activation with CsChimrson, red LED light (660 nm LED, THORLABS, USA) was applied with an intensity of 250 μW/mm$^2$ while larvae were performing sequential forward locomotion. The light intensity was measured using a laser power meter (LP1, Sanwa Electric Instrument, Japan). Trials with a duration of five seconds were applied once for each animal. The intensity of the background illumination was set as low as possible so as not to activate CsChrimson. A CCD camera (XCD-V60, SONY, Japan) under a stereo-microscope (SZX16, Olympus, Japan) with software (VFS-42, Chori imaging, Japan) was used for recording larval behavior. To quantify halt of locomotion, we counted the number of forward peristalses performed in the first two seconds after light stimulation onset in each trial. To quantify body size changes, we measured the area surrounded by the contour of the larval body in each frame (as shown in Fig. 2b).

**Imaging and quantification of muscular movements**. For the analyses of muscular movement with *Mhc > GFP*, the larvae were pinned on a silicon-filled dish and dissected along the dorsal side in TES buffered saline then the larval internal tissues were removed. In the experiment with Kir (Fig. 3a–j and Supplementary Fig. 2e, f), a lamp (X-Cite, Olympus, Japan) was used as a light source to illuminate the muscles at a light density of 7 μW/mm$^2$. To quantitate muscular contraction and relaxation, we manually measured the length of muscle 6 in segment A3, $L_{\text{raw}}(t)$, in each frame during the recordings, using an annotation tool we developed that records the $xy$ coordinate of a point clicked in a frame and calculates the distance between clicked points. The Savitzky–Golay filter was applied to the data to remove noise. We designated the frame number by $t$ and muscular length by a function of $t$, $L(t)$. $L(\tau)$ was provided as a value of $t = \tau$ of an approximate curve approximated to the data of $L_{\text{raw}}(t)$ within range $[\tau - 10, \tau + 10]$. $L(t)$ was differentiable and we defined the maximum relaxation speed (Fig. 3e, f) as the maximum time differential of $L(t)$ during muscle relaxation and maximum contraction speed (Fig. 3g, h), as the absolute value of the minimum time differential of $L(t)$ during muscle contraction. The duration of muscular contraction (Fig. 3i, j) was defined as the period from initiation of contraction to termination of relaxation.

In the experiment with CsChrimson (Supplementary Fig. 3), we used an emission splitting system (DV2 Multichannel Imaging System, Photometrics, USA) and light source (CSU-LS2WF, Solution Systems, Japan) to simultaneously perform optogenetical activation of CsChrimson and observation of GFP. Photostimulation for activation of CsChrimson was applied to larvae performing sequential peristalses. The length of muscle 6 in A3 segment during the 5 s period before and after the onset of photostimulation was measured manually.

For measurement of contraction duration of body segments in intact larvae (Fig. 3k, l and Supplementary Fig. 2c, d), larval locomotion was recorded as described above ("Optogenetics in behaving larvae"). We used blue light stimulation at a density of 450 μW/mm$^2$ to induce backward locomotion. We then manually measured contraction duration of A2 segment during forward and backward locomotion which occurred just before and after photostimulation, respectively.

**EM reconstruction using CATMAID**. Acquisition and analysis of ssTEM data are described in previous researches[27,38,45]. We manually traced axons, dendrites, presynaptic sites (identified by the presence of a T-bar) and postsynaptic sites of Canon and other neurons using CATMAID[37] (annotated in CATMAID as "A18g").

**Matching neurons identified by light microscopy and in the EM volume**. Canon neurons found by light microscopy were identified in the EM volume as follows. The lineage that a neuron of interest belongs to can be identified based on the position of its nerve entry point into the neuropile relative to landmarks. Since each lineage in the *Drosophila* larval nerve cord consists of 10–15 neurons, which are recognizable from each other by morphologies of their distal arbors or dendritic

branches, the neuron of interest can be easily identified in the lineage. In the case of Canon neurons, they were distinguished from other neurons in the lineage based on their characteristic ascending axons with collaterals in each of the two to three neuromeres anterior to the cell body and arborizing patterns of presynaptic terminals in the dorsal neuropil. In the EM dataset, we first reconstruct the low-order branches to identify Canon neurons. We then fully reconstructed the identified Canon neurons and used the reconstructed neurons as a starting point to reconstruct all their presynaptic and postsynaptic partner neurons. Analyses by light microscopy of these synaptic partner neurons were then conducted by finding sparse Gal4 and/or LexA driver lines that target them.

**Analysis of activity propagation in Canon–Canon network**. To evaluate the effect of TeTxLC expressed in Canon neurons on their activity propagation, we defined "complete" and "incomplete" peristalses waves as follows. A complete wave was defined as a wave in which Canon activities in all neuromeres examined were high ($\frac{\Delta F}{F}$ in each neuron at the peak is more than half of the maximum value of $\frac{\Delta F}{F}$ during the recording). An incomplete wave was defined as a wave in which more than one Canon neuron showed low-level neural activity ($\frac{\Delta F}{F}$ at the peak is less than half of the maximum value of $\frac{\Delta F}{F}$).

**Statistical analysis**. The two-sided Mann–Whitney U-test, the two-sided Fisher's exact test, or Chi-squared test were used to determine statistical significance as indicated in Figure legends. In multiple comparisons, the Holm method was used to avoid the family-wise error. The sample sizes are indicated within the respective figure as "n".

All statistical tests and quantification were performed using Python3. All data extracting from images and movies were performed using Fiji (Fiji is just ImageJ).

**Reporting summary**. Further information on research design is available in the Nature Research Reporting Summary linked to this article.

## Data availability
All data supporting the findings of this study are provided within the paper and its supplementary information. Source data are provided with this paper.

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

## Acknowledgements

We would like to thank Cynthia Hughes, Miki Fujioka, Bassem A. Hassan, Toshihiro Kitamoto, and Bloomington Drosophila Stock Center for providing fly lines, Herman Aberle for providing antibodies, Aref Arzan Zarin and Akira Fushiki for their contributions to EM reconstruction, Karen Hibbard for assistance with fly lines and pilot experiments, and Hiroshi Nishimaru for discussion. This work was supported by HHMI Janelia Visitor program (A.H. and S.R.P. hosted by A.C.), the Howard Hughes Medical Institute (R.D.F. and A.C.), Wellcome Trust Institutional Strategic Support Funds (105621/Z/14/Z) (S.R.P.), a Royal Society Research Grant (RG150108) (S.R.P.), and MEXT/JSPS KAKENHI grants (20K06908 and 17K07042 to H.K., 15H04255, 17H05554, 18H05113, 19H04742, and 20H05048 to A.N.).

## Author contributions

A.H. and A.N. conceived and designed the project. A.H. performed E.M. reconstruction, all experiments, and analyses, and wrote the manuscript. J.J. and H.K. performed E.M. reconstruction. S.N. identified Canon-positive Gal4 driver lines. R.D.F. generated the ssTEM data set. A.C. performed E.M. reconstruction and supervised the project. S.R.P. identified and tested Canon-specific LexA driver line and assisted with writing the paper. A.N. supervised the project and co-wrote the paper.

## Competing interests

The authors declare no competing interests.
