## [Peer Review File · Nature Communications]

Reviewer #1 (Remarks to the Author):

This study identified a new class of transtegmental excitatory premotor interneurons at the *Drosophila* larva motor circuit. Authors demonstrate by calcium imaging (GCaMP6), optogenetic activation (Chrimson), and constitutive silencing (Kir2.1) that these neurons, referred to as the canon neurons, generate prolonged longitudinal abdominal muscle relaxation that follows the peristaltic traveling wave and specifically larva crawl backwards.

Corroborating the implication of these functional studies, author's EM reconstruction placed Canon neurons downstream of the higher-order and descending command circuits for backward crawling, and upstream of a first order premotor interneuron that activates motor neurons (among many connections). Both light and electron microscopy studies show that these neuron's targets reside in anterior segments, further supporting the role of Canon neurons in transgemental inhibition to coordinate muscle contradiction that travels across segments.

These EM analyses suggest that Canon neurons at adjacent segments are heavily interconnected by chemical synapses. They found that these neurons exhibit temporally coordinated wave-like activity, which is abolished when chemical synaptic output of canon cells are blocked. Interestingly, backward peristaltic locomotion still occurs, despite being slower. Authors hypothesizes that Canon neurons may form a self-organized subcircuit to produce the wave of relaxation signals that work in concert with their previously identified inhibitory motor circuit.

A comparison of structural and functional organization across the invertebrate and vertebrate motor circuits reveals fundamental principles, but only a few systems have the capacity to study at single cell resolution. The *Drosophila* larva motor circuit is an excellent system to bridge the extremely small, and the extremely large motor circuits. This study represents clearly an original and important work, from an impressive group that seems to almost single-handedly reconstruct the larva motor circuit. I thus like to see it accepted.

But I have substantial reservation, as listed below, to recommend the acceptance at its current form. I hope that authors can address them to satisfaction so that I can full-heartedly support its publication.

My key issues with the current version of the manuscript:

1) Functional study focuses strictly on Canon neurons, whereas EM studies present Canon neurons as part of the reversal peristaltic motor circuit. It easily leave a reader unsatisfied when a paper ends with a full and complicated motor circuit model that includes all hypothetical input and output neurons of Canon, without any functional validation.

2) Functional studies of Canon neurons require further investigation. The causative roles of canon neurons came from the effect of optogenetic activation, Kirv2.1 inaction, and blocking chemical

synaptic output. The effect of chemical synapse output severely blocks calcium activity propagation among Canon neurons, but have fairly mild effect on the propagation of muscle contraction wave during peristaltic reversals (Fig. S3). This effect seems to me very different from that of silencing Canon neurons, which blocks backward peristalsis almost completely (due to shortening of intersegmental lag of contractions? See Fig. 6). There is no easy-to-find comparison from the paper – if I have interpreted these figures correctly, authors should more explicitly describe and quantify the difference, and discuss their implications.

3) At least one EM micrograph that showcases the synaptic communications among canon neurons, a key data, is not convincing to me. In Fig. 6d, for example, I am not sure if what separates A3L's T-bar and A2L is extracellular space, or extension of another neurite that extends from the lower part of the figure. If the latter, A2L cannot be a postsynaptic partner of A3L here. I always find showing consecutive sections of a synapse is needed for confident annotation, hence should also be presented similarly in publications.

4) I found the second half of the paper (EM) and the discussion of the motor circuit very hard to read through. With no background at all on the larva motor circuit in the Introduction, the anatomic detail that starts to pile up along the result sessions, and continues throughout the majority of the discussion which ends with a model, would pose obstacles for most readers. I feel authors need to restructure the paper to make it easier for a broader audience to understand, appreciate, and relate to their system and their work.

I have a few suggestions for a revision:

1) Authors should focus on getting a more complete canon neuron functional studies. To me, it is necessary to address the difference between blocking chemical synapses and silencing canon neurons. If possible (e.g. reagents are available), for authors test some predicted circuit components is functionally relevant, and whether targeted activation of canon neurons in one segment is sufficient to elicit its activity propagation, should EM-derived model become the key component of the paper (as it is now).

2) A restructure of the paper is necessary. One suggestion is to put some of the discussion in the introduction: the known and unknowns of the reversal motor circuit, if to be included in the model, without too many neuron name details, would go a long way to help readers plow through those neuron and their connections in the later part of the results and the discussion.

3) Clarify, and reduce jargons. Following are a few examples:

- GDL and iIN neurons are activated earlier than their target MNs and regulate the timing of muscular contraction rather than relaxation either via disinhibition (GDL) or by acting as a delay line (iIN).

There was really little background for this to be an easy read. I don't even know what a 'delay line' refers to here.

- Usage of segmental interneurons versus local interneurons.

I saw both terms in the paper used without definition - do they refer to the same or different type of neurons in the segments?

- Presynaptic occupation

It took me a while to figure out what it means.

Reviewer #2 (Remarks to the Author):

Hiramoto et al. present a rigorous characterization of Canon neurons, which promote muscle relaxation specifically during backwards locomotion in *Drosophila* larvae. This study combines several state-of-the-art approaches, including calcium-imaging, EM-based circuit reconstruction, and optogenetic modulation of locomotor behavior. Overall, this is an exciting study that I would be very happy to see published in Nature Communications. The work defines the neural substrates of an important component of backward movement, an ethologically relevant behavior, and extends our understanding of pre-motor circuits in a genetically tractable model organism. The statistical analyses appear appropriate, and the methods reproducible. As noted by the authors, on a more broader level this work may be relevant to the study of circuits underlying human movement disorders, which are frequently characterised by aberrant muscle co-contractions.

Below, I suggest a limited number of additional experiments that could increase the robustness of the work.

Major concerns:

1. In certain datasets, particularly Fig. 1D, the number of larvae studied appears low (e.g. $n = 2$). While I appreciate the challenging nature of these experiments, it would be preferable to see at least three independent larvae studied (ideally more, but not essential).
2. My only real concern regards the authors' postulate that Canon neurons form an autonomous pattern generator - a 'smaller pattern generator within a larger CPG' (Discussion, lines 434-435). To study the propagation of excitatory waves in Canon neurons within adjacent neuromeres, the authors express TetxLC using Canon-spGal4. However, the authors note earlier that this driver does not solely label Canon neurons, but also Wave neurons and additional neurons. This raises the possibility that reduced wave propagation might be due to inhibition of synaptic transmission from non-Canon neurons. In this light, a more comprehensive description of the Canon-spGal4 expression pattern would seem warranted. What are the 'additional neurons' and could they plausibly signal to Canon neurons? Could the authors instead use the more restricted R91C05-LexA line for these experiments? This appears more specific to Canon-neurons, and a LexAop-Kir construct for neuronal silencing has been generated by the Dickson lab. Finally, do the authors hypothesize that the Canon-Canon network truly acts as a 'smaller pattern generating circuit' i.e. can self-generate rhythmic changes in Canon network excitability? Since Canon-neurons are cholinergic, circuit motifs such as reciprocal inhibition that could drive such patterns would appear to be absent. So what is the author's model for how Canon neurons could generate patterned activity independently from the primary CPG? Any additional data that would support this premise would be useful here.

Minor concerns:

1. In Supp Fig. 5D, can the authors label the neuromeres? Although R91C05-LexA is stated to label Canon neurons in A1-A3, the GRASP signal appears to be present in more posterior segments, which was confusing to this reviewer. Could the authors also add a negative control e.g. the 1-10-GFP alone?
2. Could the authors use a more zoomed-out image for Fig. 4c? This might make the location of the active zone more clearer for the reader.

Yours sincerely,
James Jepson

Reviewer #3 (Remarks to the Author):

Regulation of coordinated muscular relaxation by a pattern-generating intersegmental circuit by Hiramoto and colleagues.

The authors have three major findings:

1. Functionally, the authors show that Canon neurons are active specifically during backward but not forward locomotion. They show that Canon neurons fire during backward peristalsis, with a phase delay relative to excitatory motor neurons of the same segment. The authors also show that Canon neurons receive a large amount of pre-synaptic input both directly and indirectly from backward “command-like” neurons from the central brain. All these data are consistent with the idea that Canon neurons play a specific role during backward locomotion.
2. Further, the authors show Canon neurons in adjacent segments are recruited in a wave like pattern. Canon neurons are cholinergic and ascend several segments towards the central brain. Canon neurons on one side of the nerve cord provide extensive input onto ipsilateral Canon neurons in other segments. Functionally, the authors find that dampening the activity of Canon neurons impairs the segmental recruitment of more posterior Canon neurons. These data support the conclusion that Canon neurons are important for intersegmental propagation of the “wave” of Canon wave activity.
3. The authors show that Canon neurons provide pre-synaptic input onto a range of inhibitory pre-motor interneurons. These pre-motor neurons synapse onto excitatory motor neurons. Consistent with this anatomical network configuration, the authors show that damping the activity of Canon neurons causes a longitudinal muscle to remain contracted for a longer duration than in controls. These data support the idea that Canon neurons are required for muscle relaxation. (The authors also show activation of Canon neurons is sufficient to halt forward locomotion.)

In general this study is a thorough and informative investigation of an important understudied topic in motor control. It provides novel circuit level insights.

Minor points:

1. Would be nice to know what exactly is targeted by the following genetic lines:
 - R91C05-GAL4 targets Canon neurons in abdominal neuromeres A3-A5 and wave neurons. What else?
 - Canon-spGAL4 targets Canon neurons in abdominal neuromeres A1-A6 and wave neurons. What else?
 - R91C05-LexA targets Canon only in A1-A3. Is that it?
2. It would be nice to confirm that Canon neurons are ChAT(+) using a second method other than immune staining, because the anti-Chat antibody is very difficult to use. For example, using ChAT RNAi to knock down Chat in Canons look at phenotypes or show the Chat staining goes away. Or use ChAT intersectional line to label Canons?

There are a number of not necessarily wrong, but not necessarily logical statements. Here is one example,

Line 140— speculates that Canons may play a role in shutting down MN activities... but actually this is counterintuitive to what one would naively think. They suggest that Canon neurons are excitatory, so one might expect they provide excitatory drive. And they suggest these neurons have presynaptic specializations in the MN output zone. All *Drosophila* MNs are excitatory. Together the naive expectation is that the neurons would cause muscle contraction, not relaxation.

Here is a second example:

Line 186- The authors they did not use the LexA line because the suitable constructs are not available. But, there has been a LexAOP-KIR2.1 line available since 2018. It has been used by some authors on this paper. (Burgos, A., Honjo, K., Ohyama, T., Qian, C.S., Shin, G.J., Gohl, D.M., Silies, M., Tracey, W.D., Zlatic, M., Cardona, A., Grueber, W.B. (2018). Nociceptive interneurons control modular motor pathways to promote escape behavior in *Drosophila*. *eLife* 7(): e26016.)

Line 157— The authors state that optogenetic experiments (loss of mobility) were consistent with the idea that Canon neurons induce relaxation of muscles. But, one could argue that the same results could be gained by contracting all muscles. Really all this result tells you is that disruption of normal canon function, impedes backward, but not forward peristalsis. This is a perfectly acceptable outcome. As is these results are over interpreted.

On the subject of behavior:

Does activation of Canon neurons also stop larvae during backward locomotion?

Are there changes in the overt body shape?

What is the effect on peristaltic locomotion when Canon neuronal activity is suppressed with KIR2.1 or TXT (or even ablation with RPR/HID)?

Line 165— Figure 3b-c, I don't really see from this data how they authors can claim that muscle relaxation is disrupted. Perhaps showing fewer events a higher resolution would help to make this point. The data as presently shown mainly show that the distribution of movement events has shifted in the experimental background.

Line 177— I would say this is the first direct evidence, not evidence that further supports the claim.

Line 193— all they can formally conclude from the Wave silencing experiment is that Canon neurons or Canon plus wave neurons are required.

Lines 255 -The fact that activation of GVLs, like that of Canon, induces muscular relaxation suggests that Canon neurons induce muscular relaxation in part by activating GVLs. Without experimental evidence, this reads as very speculative and perhaps belongs in the discussion.

Transient loss of function of GVLs is distracting. I recommend removing Figure 5J. And lines 257-261

[...]
[SEP]

To all the authors, nice work! Sorry this took me so long to review. Covid times have been crazy for me.

Reviewer #1 (Remarks to the Author):

This study identified a new class of transtegmental excitatory premotor interneurons at the *Drosophila* larva motor circuit. Authors demonstrate by calcium imaging (GCaMP6), optogenetic activation (Chrimson), and constitutive silencing (Kir2.1) that these neurons, referred to as the canon neurons, generate prolonged longitudinal abdominal muscle relaxation that follows the peristaltic traveling wave and specifically larva crawl backwards.

Corroborating the implication of these functional studies, author's EM reconstruction placed Canon neurons downstream of the higher-order and descending command circuits for backward crawling, and upstream of a first order premotor interneuron that activates motor neurons (among many connections). Both light and electron microscopy studies show that these neuron's targets reside in anterior segments, further supporting the role of Canon neurons in transgemental inhibition to coordinate muscle contradiction that travels across segments.

These EM analyses suggest that Canon neurons at adjacent segments are heavily interconnected by chemical synapses. They found that these neurons exhibit temporally coordinated wave-like activity, which is abolished when chemical synaptic output of canon cells are blocked. Interestingly, backward peristaltic locomotion still occurs, despite being slower. Authors hypothesizes that Canon neurons may form a self-organized subcircuit to produce the wave

of relaxation signals that work in concert with their previously identified inhibitory motor circuit.

A comparison of structural and functional organization across the invertebrate and vertebrate motor circuits reveals fundamental principles, but only a few systems have the capacity to study at single cell resolution. The *Drosophila* larva motor circuit is an excellent system to bridge the extremely small, and the extremely large motor circuits. This study represents clearly an original and important work, from an impressive group that seems to almost single-handedly reconstruct the larva motor circuit. I thus like to see it accepted.

But I have substantial reservation, as listed below, to recommend the acceptance at its current form. I hope that authors can address them to satisfaction so that I can full-heartedly support its publication.

My key issues with the current version of the manuscript:

1) Functional study focuses strictly on Canon neurons, whereas EM studies present Canon neurons as part of the reversal peristaltic motor circuit. It easily leave a reader unsatisfied when a paper ends with a full and complicated motor circuit model that includes all hypothetical input and output neurons of Canon, without any functional validation.

As suggested by the reviewer, we expanded functional analyses to other component neurons of the circuit (see below).

2) Functional studies of Canon neurons require further investigation. The causative roles of canon neurons came from the effect of optogenetic activation, Kirv2.1 inaction, and blocking chemical synaptic output. The effect of chemical synapse output severely blocks calcium activity propagation among Canon neurons, but have fairly mild effect on the propagation of muscle contraction wave during peristaltic reversals (Fig. S3). This effect seems to me very different from that of silencing Canon neurons, which blocks backward

pestral almost completely (due to shortening of intersegmental lag of contractions? See Fig. 6). There is no easy-to-find comparison from the paper – if I have interpreted these figures correctly, authors should more explicitly describe and quantify the difference, and discuss their implications.

We apologize that our description on this matter in the original manuscript was incomplete and misled the reviewer to think that the effect of blocking chemical synaptic outputs (TNT) and silencing (Kirv2.1) of Canon neurons on peristalses are very different. Actually, they were similar: they both did not disrupt the propagation of muscular contractions along the segments but prolonged the period of muscle contractions. We revised the manuscript to describe this point more explicitly.

3) At least one EM micrograph that showcases the synaptic communications among canon neurons, a key data, is not convincing to me. In Fig. 6d, for example, I am not sure if what separates A3L's T-bar and A2L is extracellular space, or extension of another neurite that extends from the lower part of the figure. If the latter, A2L cannot be a postsynaptic partner of A3L here. I always find showing consecutive sections of a synapse is needed for confident annotation, hence should also be presented similarly in publications.

Thank you for the suggestion. As requested, we now included consecutive EM sections showing more clearly the synaptic communications among the neurons (see Figures 4c, 5d, 6c,d).

4) I found the second half of the paper (EM) and the discussion of the motor circuit very hard to read through. With no background at all on the larva motor circuit in the Introduction, the anatomic detail that starts to pile up along the result sessions, and continues throughout the majority of the discussion which ends with a model, would pose obstacles for most readers. I feel authors need to restructure the paper to make it easier for a broader audience to understand, appreciate, and relate to their system and their work.

As suggested, we extensively restructured Introduction and Discussion to make the discussion on the motor circuit more reader friendly (see below).

I have a few suggestions for a revision:

1) Authors should focus on getting a more complete canon neuron functional studies. To me, it is necessary to address the difference between blocking chemical synapses and silencing cannon neurons. If possible (e.g. reagents are available), **for authors test some predicted circuit components is functionally relevant, and whether targeted activation of canon neurons in one segment is sufficient to elicit its activity propagation, should EM-derived model become the key component of the paper** (as it is now).

As requested, we expanded the functional analyses of the Canon circuit as much as the available genetic tools allowed us. We found Gal4 lines that specifically targets a downstream partner of Canon, A31k, and used them to show that activation of A31k neuron, like that of Canon, induces muscle relaxation (Figure 5f-h, S5i, j). This provides further support for our model that Canon regulate muscular relaxation by activating inhibitory premotor neurons.

We also tried to study whether activation of Canon neurons in one segment is sufficient to elicit its activity propagation but found it technically difficult. We generated flies that express different combinations of channelrhodopsins (such as ChR2-T159C and CsChrimson) and calcium indicators (jRGECO1b and GCaMP) in Canon neurons and tried to activate the neuron in one segment while observing the activities in other segments. However, it was difficult to obtain specific/sufficient activation and/or clear calcium signals due to complex nature of the experiments (such as lower level of expression induced by the LexA system as compared to Gal4-UAS system and light intended for calcium imaging activating channelrhodopsin). We therefore regret to say that we were unable to answer this question. We strongly feel however that a complete understanding of all aspects of Canon function is not necessary for the current

manuscript to represent a significant contribution (also please see our response to reviewer 2's major concern #2).

Finally, as mentioned above, we made clear the difference/similarity between blocking chemical synapses and silencing cannon neurons. In both cases, backward propagation of muscle contraction can occur but with prolonged contractions in each segment (as shown in the new figure, S3c, d). The purpose of the blocking chemical synapses was to study the effect of specifically inhibiting transmission but not activity of Canon neurons on the activity propagation in isolated nerve cords.

2) A restructure of the paper is necessary. One suggestion is to put some of the discussion in the introduction: the known and unknowns of the reversal motor circuit, if to be included in the model, without too many neuron name details, would go a long way to help readers plow through those neuron and their connections in the later part of the results and the discussion.

We thank the reviewer for the helpful suggestions. As suggested, we moved a section describing component neurons in reversal motor circuit from Discussion to Introduction and made the known and unknown clear in the Introduction. We also shortened the Discussion and made it more readable for people outside the field.

3) Clarify, and reduce jargons. Following are a few examples:

- GDL and iIN neurons are activated earlier than their target MNs and regulate the timing of muscular contraction rather than relaxation either via disinhibition (GDL) or by acting as a delay line (iIN).

There was really little background for this to be an easy read. I don't even know what a 'delay line' refers to here.

- Usage of segmental interneurons versus local interneurons.

I saw both terms in the paper used without definition - do they refer to the same or different type of neurons in the segments?

- Presynaptic occupation

It took me a while to figure out what it means.

Thank you for raising these points. We eliminated the use of jargons and provided appropriate definitions in the revised manuscript.

Reviewer #2 (Remarks to the Author):

Hiramoto et al. present a rigorous characterization of Canon neurons, which promote muscle relaxation specifically during backwards locomotion in *Drosophila* larvae. This study combines several state-of-the-art approaches, including calcium-imaging, EM-based circuit reconstruction, and optogenetic modulation of locomotor behavior. Overall, this is an exciting study that I would be very happy to see published in Nature Communications. The work defines the neural substrates of an important component of backward movement, an ethologically relevant behavior, and extends our understanding of pre-motor circuits in a genetically tractable model organism. The statistical analyses appear appropriate, and the methods reproducible. As noted by the authors, on a more broader level this work may be relevant to the study of circuits underlying human movement disorders, which are frequently characterised by aberrant muscle co-contractions.

Below, I suggest a limited number of additional experiments that could increase the robustnes of the work.

Major concerns:

1. In certain datasets, particularly Fig. 1D, the number of larvae studied appears low (e.g n = 2). While I appreciate the challenging nature of these experiments, it would be preferable to see at least three independent larvae studied (ideally more, but not essential).

We increased the sample number as requested.

2. My only real concern regards the authors' postulate that Canon neurons form an autonomous pattern generator - a 'smaller pattern generator within a larger CPG' (Discussion, lines 434-435). To study the propagation of excitatory waves in Canon neurons within adjacent neuromeres, the authors express TetxLC using Canon-spGal4. However, the authors note earlier that this driver does not solely label Canon neurons, but also Wave neurons and additional neurons. This raises the possibility that reduced wave propagation might be due to inhibition of synaptic transmission from non-Canon neurons. In this light, a more comprehensive description of the Canon-spGal4 expression pattern would seem warranted. What are the 'additional neurons' and could they plausibly signal to Canon neurons? Could the authors instead use the more restricted R91C05-LexA line for these experiments? This appears more specific to Canon-neurons, and a LexAop-Kir construct for neuronal silencing has been generated by the Dickson lab.

Thank you for pointing this important issue. As suggested by this reviewer and also reviewer 3, we now included more comprehensive description of the Canon-spGal4 expression pattern (see "*Drosophila melanogaster strains*" in Methods). The Gal4 targets Canon neurons, Wave neurons and putative Type II neuroblasts with no axons or dendrites. Thus, neurons whose activation and inactivation could affect larval behavior are only Canon neurons and Wave neurons. Involvement of Wave neurons can be excluded because they are not active during the fictive locomotion. Although this was reported in a previous study (Takagi et al., 2017) and was mentioned in the original manuscript, we now made this point clearer in the revised manuscript by including the data of Wave's (no) activity during fictive locomotion (Fig. S7a). We also tried to use

LexAop-TNT combined with the more restricted R91C05-LexA as suggested (please note LexAop-Kir cannot be used for this purpose since it attenuates activities in Cannon neurons). However, the experiments unfortunately did not work due to low level of TNT expression induced by R91C05-LexA (because of lower expression induced by LexA compared to Gal4 and because two transgenes, LexAop-TNT and LexAop-GCaMP6f are expressed by the same driver in this experiment).

Finally, do the authors hypothesize that the Canon-Canon network truly acts as a 'smaller pattern generating circuit' i.e can self-generate rhythmic changes in Canon network excitability? Since Canon-neurons are cholinergic, circuit motifs such as reciprocal inhibition that could drive such patterns would appear to be absent. So what is the author's model for how Canon neurons could generate patterned activity independently from the primary CPG? Any additional data that would support this premise would be useful here.

We appreciate the reviewer's thoughtful question. It is true that although our data show that Canon neurons are required for generating the proper pattern of their own activity, there is no evidence that their activation is sufficient to generate the pattern. In the original manuscript, we discussed the possibility that Canon neurons take part in generating the pattern of their own activity propagation but never insisted that their activation alone is sufficient for the pattern generation. Rather, we stated "the combination of these descending inputs, Canon-Canon signaling and some other signals from the excitatory CPGs may generate the delayed propagating activity for muscle relaxation". However, we are afraid that some expressions used in the title and text such as "pattern-generating circuits" gave the wrong impression that Canon neurons on their own generate patterned activity. We therefore revised the title and text to make these points clearer and avoid confusions. As described above in our response to reviewer 1, point 1, we also tried to study if activation of Canon neurons in one segment can induce their activity propagation but found it technically challenging. We therefore could not obtain additional data along this line.

Minor concerns:

1. In Supp Fig. 5D, can the authors label the neuromeres? Although R91C05-LexA is stated to label Canon neurons in A1-A3, the GRASP signal appears to be present in more posterior segments, which was confusing to this reviewer. Could the authors also add a negative control e.g the 1-10-GFP alone?

Thank you for pointing that appropriate control was lacking in the GRASP experiment. Since we now added new data on the function of another premotor neuron targeted by Canon neurons, A31k, in the revised manuscript, we deleted this inessential and incomplete figure in the revised manuscript.

2. Could the authors use a more zoomed-out image for Fig. 4c? This might make the location of the active zone more clearer for the reader.

As suggested, we zoomed-out the image in Fig. 4c. It indeed made the location of the active zone clearer. Thank you for the suggestion.

Yours sincerely,

James Jepson

Reviewer #3 (Remarks to the Author):

Regulation of coordinated muscular relaxation by a pattern-generating intersegmental circuit by Hiramoto and colleagues.

The authors have three major findings:

1. Functionally, the authors show that Canon neurons are active specifically during backward but not forward locomotion. They show that Canon neurons

fire during backward peristalsis, with a phase delay relative to excitatory motor neurons of the same segment. The authors also show that Canon neurons receive a large amount of pre-synaptic input both directly and indirectly from backward “command-like” neurons from the central brain. All these data are consistent with the idea that Canon neurons play a specific role during backward locomotion.

2. Further, the authors show Canon neurons in adjacent segments are recruited in a wave like pattern. Canon neurons are cholinergic and ascend several segments towards the central brain. Canon neurons on one side of the nerve cord provide extensive input onto ipsilateral Canon neurons in other segments. Functionally, the authors find that dampening the activity of Canon neurons impairs the segmental recruitment of more posterior Canon neurons. These data support the conclusion that Canon neurons are important for intersegmental propagation of the “wave” of Canon wave activity.

3. The authors show that Canon neurons provide pre-synaptic input onto a range of inhibitory pre-motor interneurons. These pre-motor neurons synapse onto excitatory motor neurons. Consistent with this anatomical network configuration, the authors show that damping the activity of Canon neurons causes a longitudinal muscle to remain contracted for a longer duration than in controls. These data support the idea that Canon neurons are required for muscle relaxation.

(The authors also show activation of Canon neurons is sufficient to halt forward locomotion.)

In general this study is a thorough and informative investigation of an important understudied topic in motor control. It provides novel circuit level insights.

Minor points:

1. Would be nice to know what exactly is targeted by the following genetic lines:

-R91C05-GAL4 targets Canon neurons in abdominal neuromeres A3-A5 and wave neurons. What else?

-Canon-spGAL4 targets Canon neurons in abdominal neuromeres A1-A6 and wave neurons. What else?

-R91C05-LexA targets Canon only in A1-A3. Is that it?

We now included in the Methods section of revised manuscript more precise description of the targets of these Gal4 lines, as requested by this and reviewer 2, as summarized below.

1. R91C05-Gal4 targets Canon neurons in abdominal neuromeres A3-A5, wave neurons, and some neurons whose cell bodies are located on lateral side of the ventral nerve cord.
2. Canon-spGAL4 targets Canon neurons in abdominal neuromeres A1-A6, wave neurons, and putative type II neuroblasts in the brain.
3. R91C05-LexA targets Canon neurons in abdominal neuromeres A1-A4 but expression could be weak or absent in some of these neuromeres.

2. It would be nice to confirm that Canon neurons are ChAT(+) using a second method other than immune staining, because the anti-Chat antibody is very difficult to use. For example, using ChAT RNAi to knock down Chat in Canons look at phenotypes or show the Chat staining goes away. Or use ChAT intersectional line to label Canons?

As suggested, we conducted the control experiments with ChAT RNAi and confirmed decreased ChAT expression in Canon neurons (Figure S1j,k).

There are a number of not necessarily wrong, but not necessarily logical statements. Here is one example,

Line 140— speculates that Canons may play a role in shutting down MN activities... but actually this is counterintuitive to what one would naively think. They suggest that Canon neurons are excitatory, so one might expect they provide excitatory drive. And they suggest these neurons have presynaptic specializations in the MN output zone. All *Drosophila* MNs are excitatory. Together the naive expectation is that the neurons would cause muscle contraction, not relaxation.

Here is an second example:

Line 186- The authors they did not use the LexA line because the suitable constructs are not available. But, there has been a LexAOP-KIR2.1 line available since 2018. It has been used by some authors on this paper. (Burgos, A., Honjo, K., Ohyama, T., Qian, C.S., Shin, G.J., Gohl, D.M., Silies, M., Tracey, W.D., Zlatic, M., Cardona, A., Grueber, W.B. (2018). Nociceptive interneurons control modular motor pathways to promote escape behavior in *Drosophila*. *eLife* 7(): e26016.)

Line 157— The authors state that optogenetic experiments (loss of mobility) were consistent with the idea that Canon neurons induce relaxation of muscles. But, one could argue that the same results could be gained by contracting all muscles. Really all this result tells you is that disruption of normal canon function, impedes backward, but not forward peristalsis. This is a perfectly acceptable outcome. As is these results are over interpreted.

We thank the reviewer for pointing these misleading and possibly illogical expressions. We corrected these statements in the revised manuscript.

On the subject of behavior:

Does activation of Canon neurons also stop larvae during backward locomotion?

We tried this but found it difficult to evaluate the effect of the optogenetic manipulation. Backward peristalses induced by head pinprick or light are not consecutive but are interspersed with pauses and turnings. It was therefore difficult to distinguish these naturally occurring stops from optogenetically induced ones.

Are there changes in the overt body shape?

We thank the reviewer for raising this important issue. We now measured the size of the larvae and showed that larval bodies are relaxed when Canon neurons are activated (Figure 2d, e). We also measure the length of muscles and found they are relaxed by the optogenetic manipulation (Figure S3).

What is the effect on peristaltic locomotion when Canon neuronal activity is suppressed with KIR2.1 or TXT (or even ablation with RPR/HID)?

As suggested, we studied the effect on peristaltic locomotion of intact larvae. When Canon neuronal activity was suppressed with Kir2.1 or TNT, duration of muscle contraction was increased during backward but not forward locomotion as was observed in the dissected larvae (Figure 3k,l; S2c, d).

Line 165— Figure 3b-c, I don't really see from this data how they authors can claim that muscle relaxation is disrupted. Perhaps showing fewer events a higher resolution would help to make this point. The data as presently shown mainly show that the distribution of movement events has shifted in the experimental background.

As suggested, we now show fewer events at a higher resolution in the revised Figure 3b-c. Prolonged muscle contractions can now be more clearly seen. Thank you for the suggestion.

Line 177— I would say this is the first direct evidence, not evidence that further supports the claim.

We corrected the text as suggested.

Line 193— all they can formally conclude from the Wave silencing experiment is that Canon neurons or Canon plus wave neurons are required.

As described above, we now added additional experiments to show that Canon neurons are required.

Lines 255 -The fact that activation of GVLs, like that of Canon, induces muscular relaxation suggests that Canon neurons induce muscular relaxation in part by activating GVLs. Without experimental evidence, this reads as very speculative and perhaps belongs in the discussion.

We apologize for overstatement. We now moved the statement to the Discussion, as suggested.

Transient loss of function of GVLs is distracting. I recommend removing Figure 5J. And lines 257-261

As recommended, we removed Figure 5j and related text in the Results and Discussion. This made the Discussion much more concise and easier to follow. We thank the reviewer for the useful suggestion.

To all the authors, nice work! Sorry this took me so long to review. Covid times have been crazy for me.

Thank you for reviewing this manuscript in this difficult time.

Reviewer #1 (Remarks to the Author):

March 29, 2021

Dear Editor and Authors,

Re: Hiramoto et al, Regulation of coordinated muscular relaxation by a pattern-regulating intersegmental circuit

Thank you for a detailed response and the clear outlining of the revision in the revised manuscript. This is a much improved version where authors have addressed all of my main concerns. To me, this version is much better organized in communicating its results to readers, with its experimental findings justifying the conclusions. I support its acceptance for publication.

I have the following minor comments and suggestions that I feel would further improve and clarify the manuscript:

1) P4. "In particular, PMSI neurons are activated." It took me a second or two to realize that PMSI is the name of one of the neurons referred to in the previous sentences. I would modify this sentence a bit to make it easier to read: "Several classes of premotor interneurons that...left-right coordination and sequential contraction of antagonistic muscles. One such class, the PMSI (perhaps spelling out the full acronym would help too) neurons..."

2) P7. Result description for Fig. 2b,c, versus Fig. 2d-e. These figure panels show that there are two phases of the effect: before 2 second, larva simply halt forward crawl, afterwards, the body gradually elongates, plateauing at about 5 sec. If this is the case, the Result section did not describe it as clearly and it can be modified to make it clearer.

3) Page 7-8. Result description for Fig. 3. I think that this is just a style issue. I would have first spelled it out while explaining result Fig. 3e that this Kir2.1 line inactivates Canon, as well other neurons (Wave), instead of leaving it at the end, as a way to justify re-running the analyses with a more specific line.

Also, two small questions for the new result using the R91C05 line: - 3K, I showed duration of contraction time. How about the effect on max. relax/contract speed (e, g)? If the effect is not restricted to the three segments that the driver expression are limited to, this may be a supportive evidence of the role of Canon-canon network.

4) Page 10-11. I appreciate the improvement made by the additional functional analyses promoted by the EM analyses. Two small suggestions: first, personally I would put Supplemental Fig. 5k in the main figure. Second, I always feel that the key challenge of the Drosophila EM analyses (if I were the one to do it) would be to confidently correlate EM profiles from a partially reconstructed map to a specific group of neurons revealed by LM studies (driver/markers). Maybe the authors can provide a bit more details in methods to help me learn a bit more on how authors to confidently identified A06, etc. A31k for their functional analyses.

5) Page 11-12. I like the additional explanation and conclusion from these experiments. I would consider however to soften the statement: I would consider the role of canon-canon connection the likely and main cause of the wave propagation. TeTx does eliminate chemical synapses to other

neurons, which may provide feedback to Canon neurons in other segments, either by proprioception, gap junction, or others. I would not be surprised that multiple factors are at play here.

6) Page 13, line 331. "patterns of spiking in excitatory and inhibitory motor neurons'. Rephrase 'spiking' to avoid confusion. Because these are non-spiking (action potentials) oscillators, statements such as 'Similarly, in the nematode *C. elegans*, phase-shifted oscillatory calcium activities in excitatory and inhibitory motor neurons regulate muscle contraction and relaxation.' would be appropriate.

Thank you for the opportunity to review this manuscript.

Reviewer #2 (Remarks to the Author):

The authors have fully addressed my concerns, and I look forward to seeing this article published in the near future. I congratulate the author's on their efforts during this difficult period.

Reviewer #3 (Remarks to the Author):

my concerns have been satisfactorily addressed.

REVIEWERS' COMMENTS

Reviewer #1 (Remarks to the Author):

I have the following minor comments and suggestions that I feel would further improve and clarify the manuscript:

1) P4. "In particular, PMSI neurons are activated.". It took me a second or two to realize that PMSI is the name of one of the neurons referred to in the previous sentences. I would modify this sentence a bit to make it easier to read: 'Several classes of premotor interneurons that...left-right coordination and sequential contraction of antagonistic muscles. One such class, the PMSI (perhaps spelling out the full acronym would help too) neurons...'

We changed the text as suggested. We agree that it is now much easier to read. Thank you for the excellent suggestion.

2) P7. Result description for Fig. 2b,c, versus Fig. 2d-e. These figure panels show that there are two phases of the effect: before 2 second, larva simply halt forward crawl, afterwards, the body gradually elongates, plateauing at about 5 sec. If this is the case, the Result section did not describe it as clearly and it can be modified to make it clearer.

We are afraid that the reviewer misread the figure and thought that time 0 in the original figure corresponded to the onset of light application. In fact, the light application was shown by a red bar and its start corresponded to 2 sec. So, halt of forward crawl and body relaxation occur simultaneously (just one phase instead of two phases of the effects). Since the figure could be misleading, we now revised it so that time 0 in the horizontal axis now corresponds to the onset of light application. We also modified the main text so that it more clearly states that halt of forward crawl and body relaxation occur simultaneously.

3) Page 7-8. Result description for Fig. 3. I think that this is just a style issue. I would have first spelled it out while explaining result Fig. 3e that this Kir2.1 line

inactivates Canon, as well other neurons (Wave), instead of leaving it at the end, as a way to justify re-running the analyses with a more specific line.

Also, two small questions for the new result using the R91C05 line: - 3K, I showed duration of contraction time. How about the effect on max. relax/contract speed (e, g)? If the effect is not restricted to the three segments that the driver expression are limited to, this may be a supportive evidence of the role of Canon-canon network.

As suggested, we now first stated while explaining result Fig. 3e that this Kir2.1 line inactivates Canon and Wave neurons in the revised manuscript. The effect on max. relax/contract speed could not be determined, since only relaxation not contraction dynamics was studied in this experiment. Contraction in intact larvae was too fast to analyze.

4) Page 10-11. I appreciate the improvement made by the additional functional analyses promoted by the EM analyses. Two small suggestions: first, personally I would put Supplemental Fig. 5k in the main figure. Second, I always feel that the key challenge of the Drosophila EM analyses (if I were the one to do it) would be to confidently correlate EM profiles from a partially reconstructed map to a specific group of neurons revealed by LM studies (driver/markers). Maybe the authors can provide a bit more details in methods to help me learn a bit more on how authors to confidently identified A06, etc. A31k for their functional analyses.

As suggested, we put Supplemental Fig. 5k in the main figure (Fig. 5i). We also provided description of how we correlated EM and LM studies in a new subsection "Matching neurons identified by light microscopy and in the EM volume" in Methods.

5) Page 11-12. I like the additional explanation and conclusion from these experiments. I would consider however to soften the statement: I would consider the role of canon-canon connection the likely and main cause of the wave propagation. TeTx does eliminate chemical synapses to other neurons, which may provide feedback to Canon neurons in other segments, either by

proprioception, gap junction, or others. I would not be surprised that multiple factors are at play here.

Thank you for the insightful suggestions. We now soften the statement as suggested by the reviewer in the revised manuscript.

6) Page 13, line 331. “patterns of spiking in excitatory and inhibitory motor neurons’. Rephrase ‘spiking’ to avoid confusion. Because these are non-spiking (action potentials) oscillators, statements such as ‘Similarly, in the nematode *C. elegans*, phase-shifted oscillatory calcium activities in excitatory and inhibitory motor neurons regulate muscle contraction and relaxation.’ would be appropriate.

We changed the text as suggested.

Thank you for the opportunity to review this manuscript.

Thank you so much for the excellent suggestions.